# Advancing Compositional Awareness in CLIP with Efficient Fine-Tuning

Amit Peleg*     Naman Deep Singh*     Matthias Hein

Tübingen AI Center, University of Tübingen

## Abstract

Vision-language models like CLIP have demonstrated remarkable zero-shot capabilities in classification and retrieval. However, these models often struggle with compositional reasoning – the ability to understand the relationships between concepts. A recent benchmark, SugarCrepe++ [11], reveals that previous works on improving compositionality have mainly improved lexical sensitivity but neglected semantic understanding. In addition, downstream retrieval performance often deteriorates, although one would expect that improving compositionality should enhance retrieval. In this work, we introduce CLIC (Compositionally-aware Learning in CLIP), a fine-tuning method based on a novel training technique combining multiple images and their associated captions. CLIC improves compositionality across architectures as well as differently pre-trained CLIP models, both in terms of lexical and semantic understanding, and achieves consistent gains in retrieval performance. This even applies to the recent CLIPS [33], which achieves SOTA retrieval performance. Nevertheless, the short fine-tuning with CLIC leads to an improvement in retrieval and to the best compositional CLIP model on SugarCrepe++. All our models and code are available at `https://clic-compositional-clip.github.io`.

## 1 Introduction

Recent advances combining multiple modalities via large models have propelled us toward systems with greater capabilities. In the vision-language domain, models like CLIP [45], BLIP [24], FLAVA [24] and ALIGN [20] have been particularly transformative, by training on *image-text* paired datasets. These Vision-Language models (VLMs) showcase not only enhanced performance on native tasks like retrieval, captioning, etc., but also enable remarkable zero-shot capabilities in image classification [26, 33], segmentation [59], human-level perception of images [13], and similar other tasks. This suggests that CLIP like models are good at associating individual concepts with images. However, several works have shown [56, 52, 11] that CLIP models struggle in understanding how concepts combine to form complex meanings, i.e., *compositionality*.

As initially highlighted in [56], these models tend to learn a "bag of words" representation of the multimodal data, making them incapable of solving simple compositional tests. For instance, when given an image of "a man in a red shirt next to a gray horse", CLIP models have been shown to choose "a man in a gray shirt next to a red horse" as the more similar of the two captions (see [11] for more such examples). To this end, several new benchmarks (WinoGround [52], VALSE [39], Crepe [36], SugarCrepe [18], SugarCrepe++ [11]) have been created to test compositionality in VLMs.

While SugarCrepe [18] mainly tests whether a model can distinguish between lexically similar but semantically different texts ($P_1$: "The *dog* is chasing a *cat*" vs. $N$: "The *cat* is chasing a *dog*"), it has been pointed out in SugarCrepe++ [11] that this is not sufficient for true understanding. Given an image of a dog chasing a cat, models should attach higher similarity of the image to both $P_1$

---

*Equal contribution. Correspondence: *amit.peleg@uni-tuebingen.de*

39th Conference on Neural Information Processing Systems (NeurIPS 2025).

and a semantically equivalent but lexically dissimilar text $P_2$ ("The cat is *being chased by* a dog") than to the wrong text, $N$. Surprisingly, it turns out that improving compositionality on Sugar-Crepe++ is non-trivial. As can be seen in Table 1, previous works perform very well on SugarCrepe (e.g., DAC [9]), but their performance on SugarCrepe++ is even worse than the pre-trained model.

Since capturing semantic similarity is crucial for the downstream performance of CLIP models, in particular retrieval, we focus on the evaluation of compositionality with SugarCrepe++.

In this paper, we introduce *Compositionally-aware Learning in CLIP (CLIC)*. This technique fine-tunes CLIP models by leveraging already available high-quality captioned datasets like PixelProse [50] or common text-image datasets like Laion [46], which we recaption using CogVLM [53]. Using a novel technique of combining images and captions allows us to generate positives as well as hard-negatives with minimal additional overhead, e.g., we do not require an LLM for generating hard-negatives [9], nor do we need to generate synthetic images [41]. In Figure 1, we show that CLIC consistently improves both on SugarCrepe++ and on image-to-text and text-to-image retrieval benchmarks compared to the pre-trained CLIP-ViT-B/32 model [45], whereas existing methods do not achieve consistent gains. To summarize, our contributions are as follows.

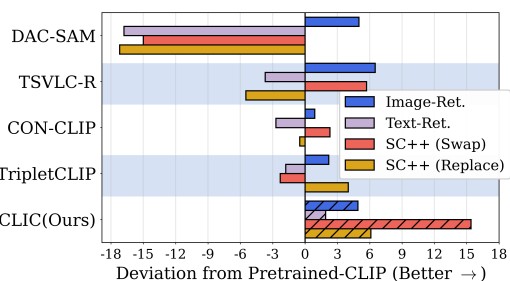

Figure 1: **Performance of fine-tuning techniques for improving compositionality** compared to the pre-trained CLIP ViT-B/32-model. Previous techniques neither yield consistent improvements on SugarCrepe++ (SC++) [11] nor for the retrieval tasks (R@5 of COCO). CLIC is the only method which shows enhanced compositionality **and** retrieval performance.

1. We propose an efficient fine-tuning method, CLIC, with little overhead, which improves compositionality and retrieval performance of pre-trained CLIP models across different sizes (ViT-B/32, ViT-B/16 and ViT-L/14) and pre-training approaches (CLIP [45], CLIPA [26], CLIPS [33]).

2. We use CLIC to improve CLIPS [33], a recent version of CLIP trained to enhance retrieval performance. Our fine-tuned model, CLIPS + CLIC, yields SOTA numbers for CLIP-like models, with a $+9\%$ average improvement on Image-to-Text (ITT) set of SugarCrepe++ and even improving on their already SOTA results in text- and image-retrieval, $(+1.3\%/+2.2\%)$.

3. Finally, we show that fine-tuning the large vision language model in LLaVA-1.5-7b [32] with CLIC vision encoder enhances it's compositional ability as measured by VQAScore [30] while maintaining its ability on tasks like question answering, captioning and chain-of-thought reasoning.

## 2 Methods and benchmarks for improving compositionality

Recent studies on VLMs have revealed significant limitations in their compositional abilities [56, 18, 52, 36, 42] via specially designed benchmarks. Throughout these works, several types of image-text models have been shown to lack compositional reasoning, ranging from those using just contrastive loss (CLIP [45], LaCLIP [12], SigLiP [57]) to the ones using different losses or architectures like CoCa [55], FLAVA [48], CLIPA [32], BLIP [16]. In this section, we introduce our notation for standard contrastive learning and provide details on methods to improve compositionality and the associated benchmarks.

### 2.1 Contrastive learning and baselines

The contrastive loss is the backbone for training VLMs like CLIP. In contrastive training, image-text paired data $(x_i, y_i)$ is embedded into a joint embedding space via the text and vision encoders. Formally, let $\psi(\cdot)$ and $\phi(\cdot)$ define the normalized embeddings of the vision- and text-encoder of the CLIP model. Then, given a batch of $m$ image-caption pairs, the model is trained using an average of image-to-text and text-to-image contrastive losses,

$$\mathcal{L}_{\text{Cont}} = -\frac{1}{2m} \sum_{i=1}^{m} \left( \log \frac{\exp(\langle \psi(x_i), \phi(y_i) \rangle)}{\sum_{j=1}^{m} \exp(\langle \psi(x_i), \phi(y_j) \rangle)} + \log \frac{\exp(\langle \psi(x_i), \phi(y_i) \rangle)}{\sum_{j=1}^{m} \exp(\langle \psi(x_j), \phi(y_i) \rangle)} \right). \quad (1)$$

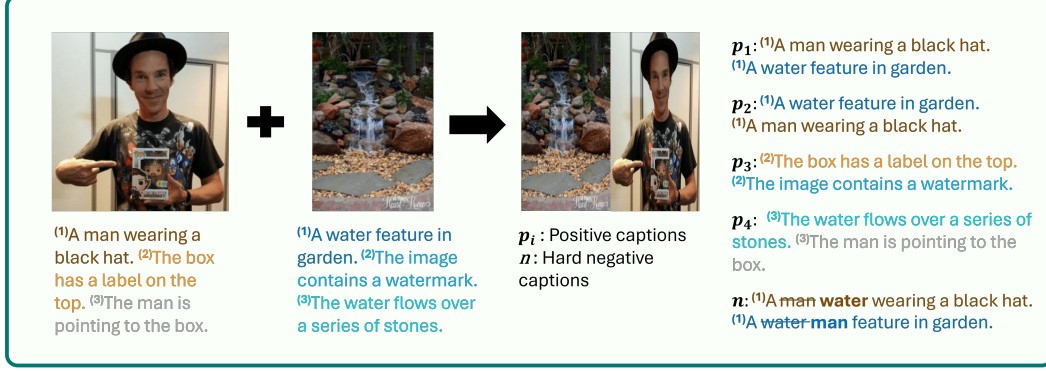

Figure 2: **Data generation scheme for CLIC**. For every image, we sample an additional image and concatenate the two. This concatenated image is the input to the model alongside five captions: $p_1$, a concatenation of the first sentence from each image. $p_2$ is a sentence-shuffled version of $p_1$. $p_3$ and $p_4$ are concatenations of two additional sentences from each caption, and $n$ is a hard negative constructed by swapping one word from each sentence of $p_1$.

Since training batches rarely include hard negative samples that challenge the "bag of words" representation, previous methods add hard negatives by slightly modifying captions to change their meaning, encouraging the model to recognize compositional cues. Denoting by $y_i^n$ the hard-negative augmentation of the caption $y_i$; methods such as NegCLIP [56] and TripletCLIP [41] adapt the image-to-text loss as follows,

$$\mathcal{L}_{\text{Neg}} = -\frac{1}{2m} \sum_{i=1}^{m} \log \left( \frac{\exp(\langle \psi(x_i), \phi(y_i) \rangle)}{\sum_{j=1}^{m} \exp(\langle \psi(x_i), \phi(y_j) \rangle) + \sum_{j=1}^{m} \exp(\langle \psi(x_i), \phi(y_j^n) \rangle)} \right). \quad (2)$$

This loss is different from the image-to-text loss used in SVLC [10] and DAC [9], which introduce an additional loss term to the CLIP loss in Eq. (1), denoted as the Single Negative (S-Neg) loss,

$$\mathcal{L}_{\text{S-Neg}} = -\frac{1}{m} \sum_{i=1}^{m} \log \left( \frac{\exp(\langle \psi(x_i), \phi(y_i) \rangle)}{\exp(\langle \psi(x_i), \phi(y_i) \rangle) + \exp\left( \langle \psi(x_i), \phi(y_i^n) \rangle \right)} \right). \quad (3)$$

### 2.2 Compositionality enhancing methods

Using hard negatives to enhance compositionality in VLMs was first introduced in NegCLIP [56], which employs spaCy [17] to swap words and phrases within a sentence. Similarly, SVLC [10] generates hard-negatives with two configurations: *(i)* SVLC-R, which uses a rule based generation with a lookup table and *(ii)* SVLC-R+L, which additionally uses an LLM based masked input completion, where masking is done for specific concepts using spaCy [17].

Dense and Aligned Captions (DAC) [9] uses the same method to create hard negatives, but first generates higher quality captions by passing the image through BLIP-2 [25]. In DAC-LLM, the enhanced captions are used as input to GPT-Neo-2.7B [14] for detailed and dense captions, and in DAC-SAM, segments of the original image, created using SAM [21], are captioned via BLIP-2. The model is then fine-tuned using a combination of $\mathcal{L}_{\text{Cont}}$ (Eq. (1)), $\mathcal{L}_{\text{S-Neg}}$ (Eq. (3)) and MIL-loss adapted from [38], which uses multiple negatives for each image.

TripletCLIP [41] uses rewritten captions from LaCLIP [12] and utilizes an LLM to generate hard-negatives by changing concepts such as objects, attributes, and relationships that change the meaning of the original caption while slightly distorting it. Afterward, a synthetic image is generated for the hard-negative captions using a text-to-image diffusion model. Finally, training is done by using a sum of image-text and text-image based $\mathcal{L}_{\text{Neg}}$ from Eq. (2).

### 2.3 Compositionality benchmarks

Several benchmarks have been proposed to measure the lack of compositionality in VLMs. These benchmarks usually add a negative caption (semantically different) and measure cross-modal Image-

to-Text (ITT) similarities across different captions. Among them is WinoGround [52], which uses two images and two corresponding captions that differ only in word order. The task of the model is to assign to each image the corresponding text and vice versa. However, it has been shown in [8] that models might fail on this benchmark for reasons other than compositionality, such as the ability to locate small and out-of-focus objects. SugarCrepe [18] measures image to text similarity and is built on the MS-COCO validation set. For each image, it requires the model to identify the correct caption from a pair of lexically similar, but semantically different captions ($P_1$ and $N$).

Recently, SugarCrepe++ [11] extended this idea by introducing $P_2$, an additional positive caption that is semantically similar to $P_1$, but lexically different from both $P_1$ and $N$. We note that SugarCrepe++ uses the additional positive caption $P_2$, for the same images, original positive captions, and plausible negative captions from SugarCrepe. To answer correctly, the model needs to assign a higher score to $P_1$ and $P_2$ compared to the score assigned to $N$. This makes SugarCrepe++ harder and more meaningful than SugarCrepe, as the model can no longer rely solely on lexical differences but has to instead understand the semantic differences between the captions. Therefore, we predominantly focus on SugarCrepe++ in this work.

On top of ITT, SugarCrepe++ also measures the uni-modal Text-to-Text (TOT) scores between the original, the semantically different, and the lexically different captions. The TOT is interesting for measuring the compositionality of the text encoder, but the ITT is arguably more important for the cross-modal downstream tasks for which CLIP models are used. Nonetheless, we report TOT for all experiments conducted in this work. Furthermore, we would also like to remark here, that many methods like [56, 49] train on MS-COCO, which may lead to knowledge leakage, as the captions and images in benchmarks like SugarCrepe and SugarCrepe++ come from the same data source.

## 3 Compositionally-aware learning in CLIP

Large-scale web-crawled datasets are known to be noisy, with many captions not reflecting their corresponding images [27, 3, 59]. This can be detrimental to compositional learning, which relies on minor modifications that alter semantic meaning. To improve the compositionality, we opt for captions that better align with the images and describe the images in detail. For this, we use approximately 1M sample subsets taken from different data sources, like, RedCaps and CC12M from PixelProse [50], and our CogVLM [53] recaptioned Laion. Details about the datasets can be found in Appendix B. We train several versions of CLIC using each dataset to demonstrate the generalizability of our approach across data sources. Additionally, since some baselines are trained using MS-COCO, for a fair comparison, we train another version on this dataset, the details of which are in Section 4.

### 3.1 Generation of positives and hard-negatives: concatenation of images and captions

To overcome the "bag of words" limitation, we aim for the model to distinguish between captions that are lexically similar but semantically different. In order to achieve this, we construct hard negatives that retain the same words in the caption but no longer correctly describe the image. For efficiency and diversity, we create these examples by concatenating pairs of images and swapping words across their two captions, ensuring that the resulting caption does not represent the concatenated images. To help the model learn that different lexical or syntactic sentences can represent the same image, we create multiple positives. For this, we use dense captions that provide varied descriptions of each image. The concatenation enables the number of combinations to grow quadratically with the dataset size, allowing CLIC to generalize better than other methods, see discussion in Appendix B.3. We detail our data preparation strategy based on concatenating *random* image-caption pairs in the following section. The process is illustrated in Figure 2 and described in Algorithm 1 in Appendix C. See Appendix B.4 for a discussion on a non-random approach.

During training, in each iteration, we sample a batch of $m$ image-caption pairs, $\{x_i, y_i\}_{i=1}^m$. We then randomly select additional $m$ pairs $\{x_i, y_i\}_{i=m+1}^{2m}$, and ensure that each selected image $x_{i+m}$ has the same orientation as $x_i$. We denote by $\text{Concat}(\cdot, \cdot)$ the process of concatenating two images or sentences and by $\text{RandomConcat}(\cdot, \cdot)$ the process of randomly shuffling the order of the two images or sentences before concatenation. We start by concatenating the images in a random order to mitigate potential biases and term the new image $u_i = \text{RandomConcat}(x_i, x_{i+m})$. Then, from the two associated captions $y_i$ and $y_{i+m}$, we create five new captions for the image $u_i$ – four positive

captions and one negative. As each original caption $y_i$ consists of several sentences, in the following, we denote by $y_i^k$ the $k_{\text{th}}$ sentence of the $i_{\text{th}}$ caption.

- $p_1$: We concatenate the first sentence of each caption and term this caption $t_i^{p_1} = \text{Concat}(y_i^1, y_{i+m}^1)$. This choice is based on the observation that the first sentence of a caption generally provides a high-level description of the image, whereas subsequent sentences describe more specific details, as demonstrated in the examples in Appendix B.1.

- $p_2$: We shuffle the order of the two sentences $t_i^{p_2} = \text{Concat}(y_{i+m}^1, y_i^1)$. We do this as both concatenated captions describe the corresponding concatenated image, irrespectively of the order of the sentences; thus, the model should be invariant to the order of the sentences.

- $p_3, p_4$: We then sample (without repetition) two additional sentences from each caption and concatenate them in a random order $t_i^{p_3} = \text{RandomConcat}(y_i^{k_1}, y_{i+m}^{k_2})$, and $t_i^{p_4} = \text{RandomConcat}(y_i^{k_3}, y_{i+m}^{k_4})$ for random indices $k_1, k_2, k_3, k_4$. This step *reinforces the model's ability to distinguish incorrect captions from correct ones*, even when the incorrect caption presents a general description of the image while the correct caption describes only a specific part. This is different from previous works, such as DAC [9], where multiple positives exist, but each caption faces its own hard-negative. An ablation on the number of positives can be found in Table 21.

- $n$: Next, we leverage the spaCy package [17] to generate hard-negatives. We decompose the first two sentences into individual words and identify their respective linguistic categories using spaCy. A table detailing these categories, along with examples, is provided in Table 7 in Appendix B. To construct hard-negatives, we randomly select a linguistic category that is common to both sentences and choose two words from that category – one from each image. If no common category exists, we randomly select a word from each image and swap them. Certain categories, such as punctuation, are excluded from this process (see Appendix B.2 for details). To prevent over-fitting to specific benchmarks, we do not impose constraints on the selection of particular categories or word positions. As a result, the generated sentence no longer accurately represents the concatenated image unless, by chance, the swapped words share the same meaning.

## 3.2 Training of CLIC

The concatenated image is fed into the model along with the five captions. We employ a combination of different loss functions, each targeting a specific desired property of the model.

**Contrastive loss:** The standard CLIP loss (1) extended to handle the four positive captions,

$$\mathcal{L}_{\text{Cont}} = -\frac{1}{8m} \sum_{i=1}^{m} \sum_{l=1}^{4} \left( \log \frac{\exp(\langle \psi(u_i), \phi(t_i^{p_l}) \rangle)}{\sum_{j=1}^{m} \exp(\langle \psi(u_i), \phi(t_j^{p_l}) \rangle)} + \log \frac{\exp(\langle \psi(u_i), \phi(t_i^{p_l}) \rangle)}{\sum_{j=1}^{m} \exp(\langle \psi(u_j), \phi(t_i^{p_l}) \rangle)} \right), \quad (4)$$

where $t_i^{p_l}$ is the positive caption of an image $u_i$ for $l \in \{1, 2, 3, 4\}$ and $i \in \{1, \dots, m\}$.

**Hard-negative loss:** We compute a hard-negative loss between each of the positive captions and the negative caption of the same image, similar to Eq. (3),

$$\mathcal{L}_{\text{S-Neg}} = -\frac{1}{4m} \sum_{i=1}^{m} \sum_{l=1}^{4} \log \left( \frac{\exp(\langle \psi(u_i), \phi(t_i^{p_l}) \rangle)}{\exp(\langle \psi(u_i), \phi(t_i^{p_l}) \rangle) + \exp(\langle \psi(u_i), \phi(t_i^{n}) \rangle)} \right), \quad (5)$$

where $t_i^n$ is the hard-negative text of $u_i$. We use a separate loss for the hard-negatives *to ensure they always influence the training and are not "masked" by other terms in the standard clip loss.*

**Uni Modal loss:** The $l_2$-distance between the first positive $t_i^{p_1}$ and its shuffled version $t_i^{p_2}$:

$$\mathcal{L}_{\text{Uni}} = \frac{1}{m} \sum_{i=1}^{m} \|\phi(t_i^{p_1}) - \phi(t_i^{p_2})\|_2 . \quad (6)$$

Since the first two positives are simply shuffled versions of each other and the concatenated images are unrelated, there is no inherent difference between the positives (except for biases in the pre-trained model). Thus, we use this loss to teach the model to be invariant to syntactic alterations that do not change the meaning of the texts. This approach differs from the text-text similarity loss used in SVLC, where the loss is between two different descriptions of the same image, one of which was generated by a language model.

Table 1: **Previous methods ITT results on SugarCrepe do not generalize to SugarCrepe++.** Comparison of methods on SugarCrepe and SugarCrepe++ compositionality benchmarks. While the improvements of previous methods on SugarCrepe do not generalize to SugarCrepe++, all versions of CLIC show strong improvements on SugarCrepe++, and also on WinoGround and SugarCrepe. Best number for each column is in **bold**, second best in underlined. Exact details on fine-tuning data for each method can be found in Appendix D.3.

| | SugarCrepe++ [11] | | | | WinoGround [52] | | | SugarCrepe [18] | | |
| | Replace | | Swap | | Text | Image | Group | Add | Replace | Swap |
| Method | ITT | TOT | ITT | TOT | Score | Score | Score | ITT | ITT | ITT |
|---|---|---|---|---|---|---|---|---|---|---|
| CLIP [45] | 69.5 | 60.5 | 45.7 | 25.9 | 31.2 | 11.0 | 8.7 | 72.9 | 80.0 | 62.7 |
| NegCLIP [56] | 70.5 | 74.1 | 56.4 | 44.8 | 30.2 | 11.0 | 8.0 | 87.6 | 85.4 | **76.5** |
| DAC-LLM [9] | 53.7 | 59.6 | 32.2 | 18.1 | 22.5 | 10.5 | 4.7 | **93.7** | **89.4** | 74.6 |
| DAC-SAM [9] | 52.3 | 60.2 | 30.7 | 18.4 | 21.2 | 12.2 | 7.5 | 91.5 | 87.0 | 73.6 |
| SVLC-R [10] | 64.0 | 69.5 | 51.4 | 28.4 | 25.2 | 8.5 | 6.7 | 85.3 | 78.9 | 68.8 |
| SVLC-R+L [10] | 61.8 | 70.0 | 45.9 | 26.4 | 28.0 | 8.2 | 5.7 | 78.4 | 75.8 | 65.4 |
| CoN-CLIP [49] | 69.0 | 70.8 | 48.0 | 32.5 | 29.5 | 10.0 | 7.2 | 83.0 | 79.7 | 65.3 |
| TripletCLIP [41] | 73.5 | 72.7 | 43.4 | 33.2 | 30.7 | 11.2 | 8.2 | 86.6 | 88.0 | 69.6 |
| CLIC-COCO | 73.5 | **79.4** | 52.9 | **47.2** | 31.0 | **13.5** | 10.0 | 85.5 | 83.8 | 69.3 |
| CLIC-LAION | 75.6 | 60.1 | 61.1 | 27.9 | 31.5 | 11.5 | 9.2 | 84.5 | 84.0 | 73.7 |
| CLIC-CC12M | 74.4 | 62.5 | 60.6 | 30.0 | 31.0 | 11.8 | 9.5 | 89.7 | 85.3 | 74.3 |
| CLIC-RedCaps | **76.0** | 57.6 | **61.5** | 23.1 | **32.2** | 12.2 | **10.7** | 86.5 | 84.8 | 72.6 |

The final objective function we use is a weighted combination of these three losses:

$$\mathcal{L} = \lambda_{\text{Cont}}\mathcal{L}_{\text{Cont}} + \lambda_{\text{S-Neg}}\mathcal{L}_{\text{S-Neg}} + \lambda_{\text{Uni}}\mathcal{L}_{\text{Uni}}, \tag{7}$$

where $\lambda_{\text{Cont}}$, $\lambda_{\text{S-Neg}}$, and $\lambda_{\text{Uni}}$ are hyperparameters controlling the contribution of each loss term. To prevent the model from deviating too much from the pretrained CLIP when training only on concatenated images, we employ the *standard clip training every second iteration*, using a single image, the first sentence, and the standard clip loss as in Eq. (1).

### 3.3 Differences between CLIC and other methods

Other approaches for generating hard text negatives differ from ours in computational cost and methodology. In DAC [9] and SVLC [10], hard-negatives tailored explicitly for specific compositionality tasks (e.g., replacing specific categories like colors, actions, etc.) are either found through look-up tables or created with computationally expensive LLM-based generation. Similarly, Triplet-CLIP uses an LLM to generate hard-negatives; for details, see Section 2.2. Closest to our work is NegCLIP, which also uses word swapping. However, NegCLIP focuses on specific word classes such as *adjectives, adverbs, verbs,* and *nouns* as described in Appendix D. We allow a swap of any two random words as long as they are from the same class, e.g., *prepositions, numbers*, and *pronouns*, as seen in Appendix B.2. Our approach offers two main advantages: cheaper computational cost compared to LLM-based generation and a design that doesn't target specific benchmarks, reducing the risk of over-fitting on existing benchmarks. Other methods also add hard-negative images, further increasing overall computational complexity. NegCLIP and CoN-CLIP search for hard images to add to the batch, and TripletCLIP generates hard images using a diffusion model. In addition, none of the methods use a combination of image pairs and corresponding text descriptions as in our method.

## 4 Experiments and evaluation

### 4.1 Experimental setup

**Dataset.** For CLIC, we fine-tune models either with CogVLM [53] recaptioned Laion images, or PixelProse [50] recaptioned images from RedCaps and CC12M (see Appendix B for details). We stress that this approach keeps our training data independent of MS-COCO, which is the basis for

Table 2: **Downstream retrieval/classification performance.** Most methods degrade zero-shot classification/retrieval performance. CLIC improves retrieval and shows minimal degradation for classification. $^*$ is not zero-shot on MS-COCO. Column-**best** and second-best are higlighted.

| Method | Fine-Tuning Data | Classification | | Text-Ret. | | Image-Ret. | |
|---|---|---|---|---|---|---|---|
| | | IMNET | ZS-10 | COCO | F30k | COCO | F30k |
| CLIP [45] | - | **63.3** | **61.4** | 74.1 | 95.1 | 54.6 | 83.5 |
| NegCLIP*[56] | MS-COCO | 60.9 | 60.7 | **83.6** | 96.4 | **72.2** | **91.7** |
| DAC-LLM [9] | CC3M | 51.1 | 54.3 | 63.3 | 79.3 | 58.1 | 88.3 |
| DAC-SAM [9] | CC3M | 52.3 | 53.9 | 57.3 | 85.6 | 58.6 | 82.2 |
| SVLC-R [10] | CC3M | 58.8 | 58.4 | 70.4 | 93.0 | 61.1 | 87.0 |
| SVLC-R+L [10] | CC3M | 59.7 | 59.0 | 68.9 | 90.4 | 61.1 | 87.0 |
| CoN-CLIP*[49] | CC3M,MS-COCO | 63.2 | 61.3 | 71.4 | 90.7 | 55.5 | 84.5 |
| TripletCLIP [41] | CC3M,CC12M | 54.8 | 52.3 | 72.3 | 91.5 | 56.8 | 85.1 |
| CLIC-COCO* | MS-COCO | 62.7 | 60.7 | 82.9 | **97.1** | 68.2 | 90.2 |
| CLIC-LAION | Laion | 61.7 | 61.0 | 75.9 | 95.0 | 60.0 | 86.7 |
| CLIC-CC12M | CC12M | 62.2 | 60.6 | 76.9 | 95.8 | 60.8 | 87.9 |
| CLIC-RedCaps | RedCaps | 62.3 | 60.2 | 76.0 | 95.6 | 59.5 | 86.3 |

benchmarks like SugarCrepe, SugarCrepe++, and retrieval evaluations. By that, we ensure CLIC always functions as a zero-shot model. However, to fairly compare with the ViT-B/32 model of NegCLIP, which is trained on MS-COCO, we also fine-tune one model on the MS-COCO [29] dataset. In our experiments, we denote each model by the respective dataset it was trained on, i.e., CLIC-COCO, CLIC-LAION, CLIC-RedCaps, or CLIC-CC12M.

**Models.** To be comparable with previous compositionality enhancing methods, we base the majority of our experiments and the ablations on ViT-B/32 models pre-trained by OpenAI [45]. However, to highlight the adaptability and scalability of our approach, we also fine-tune other commonly used pre-trained architectures, such as ViT-B/16 and ViT-L/14 from [45]. Up to our knowledge, among methods improving compositionality, only CoN-CLIP [49] provides checkpoints for these architectures. Additionally, we fine-tune more recent ViT-L equivalent VLMs, including CLIPS and CLIPA to show that our method can enhance compositionality even in models that already possess compositional abilities and are pre-trained on varying datasets such as Re-DataComp (CLIPS), WIT (CLIP) and DataComp (CLIPA). More model specific details can be found in Appendix D.1.

**Training details.** In all of our experiments for CLIC, we keep the vision encoder frozen and fine-tune only the text encoder at an image resolution of $224 \times 224$. An ablation showing that this works better than fine-tuning the entire model can be found in Table 20 in the appendix. Only for experiments associated with LLaVA [32], we fine-tune the complete model at a higher resolution of $336 \times 336$. We defer other training details like LR, loss parameters in Eq. (7), etc, and a discussion on our NegCLIP baselines (denoted everywhere with a $^\dagger$), used whenever no checkpoints are available to Appendix D.

**Evaluation.** We evaluated the models on the compositionality benchmarks SugarCrepe++ [11], WinoGround [52], and SugarCrepe [18]. Additionally, we assess the effects of fine-tuning on downstream tasks such as classification on IMAGENET [6] and on ZS-10, which computes the average score across ten standard classification datasets, detailed in Appendix D.2. In addition, we report image $(T \rightarrow I)$ and text retrieval $(I \rightarrow T)$ Recall@5 scores on MS-COCO (Val2017) [29] and Flickr30k [43]. We show Recall@1 numbers for retrieval and detailed SugarCrepe and SugarCrepe++ numbers for all the models in Appendix E.

### 4.2 Comparing to other methods

Given that most publicly available checkpoints of compositionality methods use ViT-B/32 based CLIP, we first focus on this architecture. From Table 1, we observe that most methods perform worse than the pre-trained CLIP on SugarCrepe++. Notably, only NegCLIP, CoN-CLIP, and the CLIC models improve or are on par with the pre-trained baseline. Among these, CLIC-COCO is better than NegCLIP and CoN-CLIP (all three were trained on MS-COCO), while CLIC-RedCaps

Table 3: **CLIC improves different Large (ViT-L/14) CLIP models.** CLIC improves several versions of CLIP, across architectures and pre-training datasets.* is not zero-shot for MS-COCO evaluations. WinoGround results for these models are in Table 27. Detailed retrieval, SugarCrepe++, SugarCrepe results and more architectures can be found in Appendix E.

| Method | Downstream Evaluations | | | | Sugarcrepe++ | | | | Sugarcrepe | | |
| | Classification | | COCO Ret. | | Rep. | | Swap | | Add | Rep. | Swap |
| | IMNET | ZS-10 | $I \to T$ | $T \to I$ | ITT | TOT | ITT | TOT | ITT | ITT | ITT |
|---|---|---|---|---|---|---|---|---|---|---|---|
| CLIP [45] | 75.5 | 72.3 | 79.0 | 60.0 | 70.7 | 58.7 | 44.7 | 22.7 | 74.9 | 79.5 | 61.4 |
| NegCLIP† | 73.9 | 70.6 | 77.7 | 65.9 | 70.6 | 63.5 | 50.5 | 35.8 | 83.6 | 82.8 | 70.0 |
| CoN-CLIP*[49] | 75.9 | 72.4 | 78.4 | 63.1 | 72.2 | 71.3 | 46.0 | 30.5 | 83.9 | 81.3 | 64.2 |
| CLIC-LAION | 74.2 | 71.8 | 80.1 | 65.4 | 79.0 | 59.1 | 59.8 | 26.3 | 87.7 | 85.6 | 71.3 |
| CLIC-RedCaps | 75.2 | 72.1 | 81.3 | 63.9 | 76.0 | 55.4 | 55.2 | 20.0 | 84.3 | 83.8 | 66.1 |
| CLIP-A [26] | **79.6** | **78.3** | 85.4 | 70.5 | 74.8 | **74.8** | 45.8 | 30.8 | 85.8 | 82.9 | 63.8 |
| CLIC-LAION | 78.4 | 78.0 | 84.8 | 71.5 | 78.4 | 66.3 | 55.1 | 30.7 | 92.9 | 88.2 | 71.6 |
| CLIC-RedCaps | 78.9 | **78.3** | 85.2 | 71.2 | 80.2 | 62.8 | 61.2 | 22.8 | 93.4 | 85.9 | 71.5 |
| CLIPS [33] | 78.4 | 76.5 | 90.4 | 77.3 | 79.3 | 74.2 | 60.6 | **51.1** | 89.4 | 88.0 | 79.6 |
| CLIC-LAION | 76.7 | 76.2 | 90.4 | 79.4 | 84.3 | 71.9 | 73.0 | 49.4 | **96.7** | **92.4** | **85.7** |
| CLIC-RedCaps | 77.2 | 76.6 | **91.7** | **79.5** | **84.9** | 71.4 | **75.1** | 50.2 | 95.3 | 91.1 | 84.5 |

has lower TOT with a higher ITT (10.2% gain on pre-trained model across *replace* and *swap*). We also evaluate all models on the WinoGround benchmark, where, apart from CLIC, no other method improves on the pre-trained model, highlighting the effectiveness of CLIC. Moreover, CLIC improves in compositionality regardless of the fine-tuning dataset, showing that CLIC works for different sources of captions and images used for fine-tuning. An interesting finding in Table 1 is that many models perform well on ITT in SugarCrepe but struggle on SugarCrepe++ ITT, an extended version including lexically different but semantically similar positives. This stark difference for methods like DAC (achieves 89.4% on SugarCrepe *replace* ITT but only 53.7% on SugarCrepe++) suggests these models detect lexical changes rather than truly learning compositional relations. This likely stems from their over reliance on hard-negatives tuned for previous benchmarks (see Appendix D.4).

**Downstream tasks.** We evaluate models on standard retrieval benchmarks (MS-COCO, Flickr30k) and classification tasks (IMAGENET, ZS-10) in Table 2. While TripletCLIP performs well on SugarCrepe++ and WinoGround, it exhibits significant degradation on IMAGENET and ZS-10. In contrast, our CLIC models show minimal performance degradation on classification tasks, with CLIC-CC12M achieving the highest text retrieval performance among models not trained on MS-COCO while also improving image retrieval over the pre-trained baseline.

### 4.3 Ablation Study for CLIC

In Table 4 we ablate the components of CLIC in steps (C1-C5) and present the analogous non-concatenated, single image version of CLIC (B1-B4). The CLIC baseline (C1) is fine-tuning on the concatenated images with caption $p_1$ and in (C2) we add our single hard negative. These two steps are sufficient to improve on SugarCrepe but not on SugarCrepe++. This suggests that the hard-negative distractor helps the model to detect lexical changes, but not to distinguish between semantically similar sentences and ones with different meaning. Adding multiple positives (C3) and the uni-modal loss (C4) improves significantly on SugarCrepe++ while performance on SugarCrepe slightly reduces. Thus, adding diverse positives and enforcing invariance of semantically identical captions lets the model identify semantically similar captions while being sensitive to lexically similar but semantically dissimilar captions. In the last step (C5) we add the iterates with single images for standard contrastive loss, which further improves compositionality and downstream task performance by eliminating the potential concatenated image biases. The non-concatenated, single-image variant of CLIC shows similar but smaller improvements in compositionality while being significantly worse in retrieval. Here, the NegCLIP† variant (similar to B2, but employing $\mathcal{L}_{\text{Neg}}$ (2) in place of $\mathcal{L}_{\text{S-Neg}}$ (3)) was used to compare CLIC to NegCLIP on a dataset other than MS-COCO, in order to avoid potential test data leakage from MS-COCO during training, as discussed in Section 2.3.

Table 4: **Effects of different components in CLIC.** The non-concat baselines represent single-image versions of CLIC. Rows : standard fine-tuning on our 1M Laion set with either a concatenated or a single image and a single caption. Rows present standard one hard negative training. In the single image case, with either the $\mathcal{L}_{\text{Neg}}$ (2) for **NegCLIP**$^\dagger$ or the $\mathcal{L}_{\text{S-Neg}}$ (3) for $B2$. Finally, rows feature CLIC (with all the components as presented in Section 3) with its analogous single-image version.

| | Method | Downstream Evaluations | | | | Sugarcrepe++ | | | | Sugarcrepe | | |
|---|---|---|---|---|---|---|---|---|---|---|---|---|
| | | Classification | | COCO Ret. | | Rep. | | Swap | | Add | Rep. | Swap |
| | | IMNET | ZS-10 | Text | Image | ITT | TOT | ITT | TOT | ITT | ITT | ITT |
| $A1$ | CLIP [45] | 63.3 | 61.4 | 74.1 | 54.6 | 69.5 | 60.5 | 45.7 | 25.9 | 72.9 | 80.0 | 62.7 |
| | **NegCLIP**$^\dagger$ | 61.0 | 60.5 | 72.0 | 59.7 | 67.9 | 64.7 | 54.5 | 36.1 | 82.4 | 80.9 | 70.7 |
| **Single image Baselines** (Non-Concat) | | | | | | | | | | | | |
| $B1$ | Fine-tuned | 62.1 | 61.8 | 76.4 | 60.4 | 69.6 | 69.2 | 49.8 | 35.9 | 83.4 | 80.7 | 68.8 |
| $B2$ | + Single hard-negative | 60.6 | 59.8 | 70.4 | 57.9 | 66.6 | 63.8 | 55.0 | 36.2 | 81.9 | 80.5 | 70.6 |
| $B3$ | + Multi Positives | 58.5 | 59.2 | 58.2 | 56.6 | 70.2 | 48.5 | 58.5 | 23.1 | 77.7 | 76.0 | 64.7 |
| $B4$ | Single-Img Baseline | 61.3 | 60.6 | 67.3 | 58.0 | 74.1 | 53.9 | 60.2 | 27.8 | 81.0 | 79.9 | 69.7 |
| **Ablating CLIC** (Concat) | | | | | | | | | | | | |
| $C1$ | Fine-tuned Concat | 62.6 | 61.2 | 76.5 | 59.1 | 69.8 | 66.7 | 49.3 | 35.9 | 82.7 | 81.9 | 68.2 |
| $C2$ | +Single hard-negative | 60.9 | 60.0 | 73.9 | 57.8 | 68.7 | 63.6 | 53.7 | 34.2 | 84.2 | 83.8 | 73.2 |
| $C3$ | +Multi Positives | 61.3 | 59.6 | 61.3 | 54.8 | 74.2 | 47.4 | 59.0 | 19.5 | 84.2 | 81.9 | 70.9 |
| $C4$ | +Uni-Modal Loss | 60.9 | 59.3 | 59.9 | 54.5 | 74.2 | 50.7 | 59.1 | 20.3 | 83.4 | 81.6 | 70.3 |
| $C5$ | **+CLIP iterate (CLIC)** | 61.7 | 61.0 | 75.9 | 60.0 | 75.6 | 60.1 | 61.1 | 27.9 | 84.5 | 84.0 | 73.7 |

Table 5: **Comparison to SOTA models for SugarCrepe++ and WinoGround.** CLIPS fine-tuned with CLIC-RedCaps attains on average the strongest compositionally aware model, especially among CLIP variants, as seen on both SugarCrepe++ and WinoGround.

| Model | Params (M) | SugarCrepe++ | | | | | | WinoGround | | |
|---|---|---|---|---|---|---|---|---|---|---|
| | | Replace | | Swap | | Average | | Score | | |
| | | ITT | TOT | ITT | TOT | ITT | TOT | Text | Image | Group |
| FLAVA [48] | 358 | 74.4 | **75.0** | 56.8 | **51.5** | 67.3 | **65.5** | 32.2 | **20.5** | 14.2 |
| BLIP-Base [24] | 225 | 73.8 | 73.9 | 54.0 | 42.1 | 66.1 | 61.1 | 35.8 | 15.8 | 13.3 |
| BLIP-Large [24] | 470 | 81.7 | - | 59.8 | - | 72.9 | – | 37.7 | 13.5 | 10.5 |
| ViT-H/14 [46] | 986 | 73.8 | 72.1 | 48.8 | 39.1 | 63.7 | 58.8 | 33.5 | 12.7 | 10.5 |
| ViT-bigG/14 [46] | 2540 | 76.6 | 73.2 | 51.5 | 40.2 | 66.6 | 59.7 | 35.5 | 15.0 | 12.0 |
| CLIPS [33] | 427 | 79.3 | 74.2 | 60.6 | 51.1 | 71.9 | 64.6 | 36.2 | 16.0 | 12.5 |
| CLIPS + CLIC | 427 | **84.9** | 71.4 | **75.1** | 50.2 | **81.0** | 62.9 | **41.7** | 17.5 | **15.2** |

## 4.4 Generalization to larger architectures and models

Next, we measure the generalization of the method to larger architectures such as ViT-L/14 across models pre-trained on different datasets (CLIPA and CLIPS). Results on ViT-B/16 can be found in Table 24 in the appendix. We fine-tune each of the pre-trained models, CLIP, CLIPA, and CLIPS, on our 1M Laion subset and on RedCaps (PixelProse) subset using CLIC. As most other compositionality enhancing methods do not explore larger architectures, the only baselines we have are CoN-CLIP and our NegCLIP$^\dagger$ (the details of which can be found in Appendix D). In Table 3, we can see that for all types of pre-trained models, CLIC-RedCaps shows consistent improvements on the ITT score for SugarCrepe++: +7.4% for CLIP, +9.4% for CLIPA, and +9.2% for CLIPS when averaged across *replace* and *swap*. These improvements are on a similar scale as the ones for the ViT-B/32 model, suggesting that even for larger architectures, CLIC can attain substantial gains.

What we find even more remarkable are the improvements for image and text retrieval attained by CLIC. Specifically, for CLIPS (which has to the best of our knowledge SOTA retrieval numbers of a CLIP model), CLIC further improves text retrieval by 1.3% and image retrieval by 2.2% on

MS-COCO. These gains come at a tiny fraction (0.01%) of the pre-training cost, as we only see 850k (RedCaps) resp. 1M (Laion) samples compared to 13B samples seen during pre-training of CLIPS.

**Constrasting again SOTA.** In Table 5, we pitch our best model in CLIPS + CLIC against the SOTA CLIP like and other models on SugarCrepe++. To the best of our knowledge, **CLIPS + CLIC attains the best available ITT numbers on SugarCrepe++ across all architectures.** We note that CLIPS in itself is already good for ITT on SugarCrepe++ but our short fine-tuning enhances it significantly further in terms of compositional understanding. For TOT, among CLIP like models CLIPS + CLIC performs similarly to the best model (CLIPS) in this category. We were unable to evaluate BLIP-Large for TOT, since the individual BLIP-Large encoders are not publicly available. The WinoGround results for CLIPS + CLIC are the best in text and group score, while the image score is the best among CLIP-like models. As far as we know, CLIPS + CLIC attains SOTA WinoGround scores among CLIP based models.

### 4.5 CLIC vision encoder in LLaVA

Unlike our previous experiments, we fine-tune the CLIP model with CLIC at $336 \times 336$ resolution while unfreezing both the vision and text encoders. We then replace the CLIP vision encoder in LLaVA-1.5-7b (Vicuna) [31] with our CLIC version and fine-tune the full LLaVA model following standard projector pre-training (full details in Appendix D.5). As shown in Table 6, CLIC enabled LLaVA-1.5 achieves notable compositional reasoning improvements. For SugarCrepe++, we observe gains across most categories with marginal degradation for swap object, while WinoGround scores improve for text and group sets by more than 6% and 3% respectively. Crucially, these gains preserve performance on standard VLM benchmarks (GQA, TextVQA, SQA-I), with MME (perception) improving from 1441.0 to 1465.3. These results show that even short compositional fine-tuning via CLIC can significantly enhance fine-grained visual-linguistic understanding while maintaining general vision-language capabilities, as further illustrated by qualitative examples in Figure 6.

Table 6: **Comparing standard LLaVA-1.5-7b (CLIP) with it's CLIC enabled version.** Using the VQAScore metric from [30], we evaluate compositionality via SugarCrepe++ (ITT) and WinoGround, while showing results for some benchmarks (VLM-tasks) native to LLaVA. Across compositionality tasks, CLIC enabled LLaVA-1.5 is almost always better than CLIP based LLaVA while maintaining the same performance on nominal VLM tasks. Task-wise best compositional model is **highlighted**.

| Model | SugarCrepe++ (ITT) | | | | | WinoGround | | | VLM-tasks | | | |
|---|---|---|---|---|---|---|---|---|---|---|---|---|
| | Replace | | | Swap | | | | | | | | |
| | Obj | Att | Rel | Obj | Att | Text | Img | Grp | GQA | TextVQA | SQA-I | MME (P) |
| LLaVA-1.5 (CLIP) | **94.3** | 80.4 | 73.3 | **66.9** | 69.8 | 36.7 | 38.2 | 23.2 | 60.6 | 57.1 | 69.2 | 1441.0 |
| LLaVA-1.5 (CLIC) | **94.3** | **80.8** | **73.8** | 66.5 | **71.2** | **43.5** | **42.0** | **26.2** | 62.1 | 57.7 | 68.2 | 1465.3 |

## 5 Conclusion

We introduce CLIC, a simple, low-cost, and scalable method for boosting compositionality in vision-language models, as measured by SugarCrepe++. Notably, this improvement comes with a minimal degradation in downstream tasks like zero-shot classification and enhanced text and image retrieval performance. CLIC works across different good caption quality datasets. We further illustrate the generalization of CLIC to variedly (dataset/training-scheme) pre-trained models, architectures. Lastly, we show that CLIC even improves performance on models already exhibiting strong compositional and retrieval abilities like CLIPS and yields SOTA results on SugarCrepe++ as well as image retrieval. From our initial experiments, CLIC powered LLaVA improves in compositional reasoning over the standard version. We believe compositionally aware vision encoders can further help large VLMs like LLaVA on more tasks, a detailed study of this is left for future work.

## Acknowledgements

The authors thank the International Max Planck Research School for Intelligent Systems (IMPRS-IS) for supporting AP and NDS. We acknowledge support from the Deutsche Forschungsgemeinschaft (DFG, German Research Foundation) under Germany's Excellence Strategy (EXC number 2064/1, project number 390727645), as well as in the priority program SPP 2298, project number 464101476. We are also thankful for the support of Open Philanthropy and the Center for AI Safety Compute Cluster. Any opinions, findings, and conclusions or recommendations expressed in this material are those of the author(s) and do not necessarily reflect the views of the sponsors. We thank the reviewers for the discussions and suggesting additional experiments.

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

# Contents

## A  Limitations

Our work considers only a fraction of contrastive models, we believe CLIC should work similarly for other models like SigLIP [57], which we were unable to test due to compute and time constraints. Moreover, we were only able to test LLaVA [31], whereas other large VLMs like VILA [28] and CogVLM [53] might also benefit from CLIC, which we leave for future work.

## B  Data preparation

In this section, we describe the data preparation process in detail. Appendix B.1 showcases randomly selected images and captions from the PixelProse [50] dataset and from the Laion [46] dataset, along with captions generated for these images by CogVLM [53]. This part also presents further details on these datasets. In Appendix B.2, we explain the hard-negative generation procedure. In Appendix B.3 we show examples of the limitations of using a single image for generating negatives as done in [56]. Appendix B.4 shows ablations regarding other ways of combining images for CLIC.

### B.1  Training datasets

In this section, we present our data generation process. To show that CLIC performs well across diverse datasets, we train models from three different sources. We took a subset of 1M images from Laion with captions that were generated by CogVLM [53], using the query presented in Figure 3. We also used a recently proposed high-quality caption dataset in PixelProse [50], which contains 850k available samples from the `RedCaps` subset. In order to compare faithfully to our baselines in Table 1 who use CC3M, we also use a 1.2M subset of CC12M that comes from the same data stream as CC3M to train a ViT-B/32 model. Since our method requires detailed captions, we use the recaptioned version of CC12M from PixelProse. These two datasets come from a different stream of internet data than Laion. A selection of random image-caption pairs from both kinds of datasets is provided in Figure 4. As can be seen from the examples, the first sentence in each caption usually provides a general description of the image, while the other part of the caption often captures more specific details. To clean the data, we removed the starting strings like "This picture depicts/shows/demonstrates:". Then, since the generated captions tend to be longer than those in the standard datasets like Laion and DataComp (the average caption length in our recaptioned dataset is 78 words), we split the captions into individual sentences as done in [33, 59]. This results in an average sentence length of 16 words for captions generated by CogVLM [53]. We retain only images whose captions contain more than one sentence, which is more than 99% of the 1M set for Laion. This approach favors captions that provide both general context and specific details about the images. For a fair comparison with NegCLIP, we also train CLIC with MS-COCO, the details of which are in Section 4.

> **Query:** Can you please describe this image in a long and detailed paragraph? Please keep your descriptions factual and terse but complete. DO NOT add any unnecessary speculation about the things that are not part of the image. The description should be purely factual, with no subjective speculation.

Figure 3: **Query used for re-captioning for the CogVLM model [53].**

Table 7: **List of Universal POS Tags with their Meaning and Examples.** Examples are taken from ChatGPT. For further information, refer to https://universaldependencies.org/u/pos/.

| POS Tag | Meaning | Example Words |
|---------|---------|---------------|
| **ADJ** | Adjective | happy, blue, large |
| **ADP** | Adposition (preposition) | in, on, under, with |
| **ADV** | Adverb | quickly, very, well |
| **AUX** | Auxiliary verb | is, was, will, do |
| **CCONJ** | Coordinating conjunction | and, but, or |
| **DET** | Determiner | the, a, an, this |
| **INTJ** | Interjection | wow, oh, hey |
| **NOUN** | Noun (common & proper) | dog, city, John |
| **NUM** | Numeral | one, two, 100 |
| **PART** | Particle | not, 's (as in John's) |
| **PRON** | Pronoun | he, she, it, they |
| **PROPN** | Proper noun | London, Microsoft |
| **PUNCT** | Punctuation | ., !, ? |
| **SCONJ** | Subordinating conjunction | because, although, if |
| **SYM** | Symbol | $, %, & |
| **VERB** | Verb | run, eat, write |
| **X** | Other (foreign words, typos) | jdhfk, asdfg |

## B.2 Hard negative creation

Our hard negative creation process utilizes the spaCy package to identify the part-of-speech (POS) of words in a sentence. We consider only the coarse-grained POS level, which is universal and not language-specific. The complete list of universal POS tags is provided in Table 7. For each sentence, we classify all words, group them by their POS, and determine eligible categories for swapping. A valid swap nomination requires at least two words from the same POS category, excluding the following: "AUX", "CCONJ", "DET", "INTJ", "PART", "PUNCT", "SCONJ", "SYM" and "X". Once a valid category is identified, we randomly select it and swap two randomly chosen words within that category. We deliberately did not select specific tags to avoid overfitting to current compositionally benchmarks. If no category falls into these cases, we swap two words at random.

## B.3 In-scene vs across-scene swaps

Creating negatives by swapping words between sentences from different scenes offers several advantages over swapping within a single scene:

1. Swapping words within a single sentence can lead to negatives due to the changes in word order. For example, the sentence from NegCLIP: "The horse is eating the grass" can become "The grass is eating the horse". However, swapping words across sentences from different images greatly increases the diversity of negative examples. Taking the example from the paper (Section 1), concatenating the "The horse is eating the grass" image with an image of a dog chasing a cat may lead to the new negative: "The horse is chasing the cat. The dog is eating the grass". Since each image can be paired with each of the other images, this scales the diversity of negatives quadratically and helps the generalization of the method.

2. Creating negatives from a single scene, like in NegCLIP, may result in a modified sentence that still accurately describes the image. This might happen as in many sentences, changing the order of the words does not change the semantic meaning of the sentence. This is different from changing words across scenes, where the concatenated sentence will reflect the concatenated image only if we by chance swap words that have the same meaning (*we remind the reader that we do not allow swaps of the same word, i.e., 'man' can not be swapped with 'man'*). Here are a few examples from the CC12M subset from PixelProse:

   • "In the top left corner, a person wearing a green jacket" – switching *left* and *top* does not change the meaning.

- "there is a tuna salad with celery, onion, and mayonnaise" – changing *onion* and *celery* does not change the meaning.
- "Bugs Bunny, Taz, Lola Bunny, and Daffy Duck are standing on either side" – changing *Lola* and *Bugs*.
- "The Jeep has a black bumper, black wheels" – changing *bumper* and *wheels*.
- "A dining room and living room" – changing *dining* and *living*.
- "A black and gray backpack" – changing *gray* and *black*.

## B.4  Non-random image concatenation

Combining images at random offers several advantages, such as computational efficiency and reduced risk of overfitting to specific benchmarks. However, to examine whether generating more plausible or grammatically correct negatives leads to better results (at the expense of higher computational resources), we construct negatives in a non-random manner using two strategies:

1. Combining images that share a common noun.
2. Swapping words with different semantic meanings and selecting images accordingly.

**Common nouns**

We begin by extracting all the nouns from the first sentence of each image using spaCy. For each image, we then sample up to five other images (without repetition) that share a common noun and use this to concatenate the images. We then continue as usual by swapping random words (excluding the common noun). Although we witnessed a higher $S_{neg}$ loss (Eq. (5)), indicating that the hard negatives are indeed harder, the results on the downstream tasks remained largely unchanged (see Table 8, row SAME NOUN).

**LLM swaps**

To test the impact of non-meaningful or ungrammatical negatives, we made the following modification to our method to construct negatives that are grammatically correct by leveraging LLM based swaps:

1. We extracted the fine-grained Part-of-Speech (POS) tags for each word in the first sentence using spaCy's most computationally expensive model, which took around 5 hours. In contrast, the version used for the random image-caption pairs employed a more computationally efficient model and took only a few minutes.

2. For each word, we queried Gemini 2.5-flash [5] to generate up to five replacement candidates from the same POS category, such that replacing the word changes the meaning of the caption, while keeping the sentence coherent. **Note:** these queries to Gemini via API calls took over 24 hours of compute time to find word replacements for the 1M Laion subset. In comparison, fine-tuning a CLIP model with CLIC only takes around 40 minutes.

3. During training, for each caption, we randomly selected a POS category and then randomly chose a word from that category. The second image and word replacement were selected based on the candidates from step 2.

This significantly more expensive variant of CLIC (LLM SWAPS in Table 8) results in harder negatives (as evidenced by a higher loss during training) and leads to improvements in Replace (ITT) for SugarCrepe++ but degrades Swap (ITT) and TOT. While at first surprising, we hypothesize that the lower diversity of negatives compared to our original computationally cheap version of CLIC, with random replacement of words from the same POS category, could be a reason for this.

To check the quality of the negatives, we hand-labelled 75 negative samples from both LLM SWAPS and the random negatives used to train **CLIC-LAION**. This process yielded 69% meaningful, grammatically correct negatives with distinct semantic meanings from LLM SWAPS, compared to 28% from the **CLIC-LAION**. We include below 5 coherent and grammatically correct hard negatives examples, and 5 nonsensical or grammatically incorrect ones. While there are some errors, the overall quality is sufficient to prevent the model from relying solely on shortcuts, as many examples remain meaningful and grammatically correct, and the errors are relatively subtle.

Correct hard negative examples:

(postcard ↔ telegram)

N: a collage of various telegram marketing materials. a vintage postcard.

(stripes ↔ swirls)

N: a black satin jacket with white swirls on the collar and cuffs. a graphic design element that features a large red heart at the center surrounded by intricate golden stripes and patterns

(short ↔ thick)

N: a portrait of a woman with thick textured hair that has a mix of brown and blonde colors. an indoor setting possibly a café or a bakery with a man pouring a short brown liquid which appears to be chocolate onto a white countertop.

(plain ↔ fancy)

N: a woman standing against a fancy background. a scrapbooking paper pack titled 'sunny memories' by plain pants designs.

(corners ↔ hearts)

N: a pair of red envelopes with intricate gold embroidery on the hearts. two wooden objects that are shaped like corners.

Nonsensical or grammatically incorrect samples:

(whistle ↔ shout)

N: a red kettle with a shout attached to its spout. a man in a dramatic pose seemingly in the middle of a punch or a whistle

(made ↔ destroyed)

N: a hand-destroyed card with a floral design. a scene of destruction with a building that appears to have been damaged or made.

(printed ↔ removed)

N: a black bag with white lines and the word 'thule' removed on it. one of the nuts is shown with its cap printed revealing the coiled thread inside.

(setting ↔ breaking)

N: this image captures a moment in an outdoor park breaking where two individuals are engaged in a playful activity. a bear's face that appears to be setting through a shattered glass surface.

(appears ↔ disappears)

N: it disappears to be a motorized treadmill with a digital display on the top. the headline reads 'german billionaire appears on matterhorn' and it is dated april 11 2018 at 11:35 am cdt.

Table 8: **Comparing with non-random image concatenation.** All methods use LAION data captioned by CogVLM and the ViT-B/32 architecture. Although training with non-random image concatenation leads to higher $\mathcal{L}_{\text{S-Neg}}$ loss (Eq. (5)), the performance of the model remains the same.

| | Downstream Evaluations | | | | | | SugarCrepe++ | | | | SugarCrepe | | |
| | Classification | | Text Ret. | | Image Ret. | | Replace | | Swap | | Add | Rep. | Swap |
| Method | IMNET | ZS-10 | COCO | F30k | COCO | F30k | ITT | TOT | ITT | TOT | ITT | ITT | ITT |
|---|---|---|---|---|---|---|---|---|---|---|---|---|---|
| CLIP [45] | 63.3 | 61.4 | 74.1 | 95.1 | 54.6 | 83.5 | 69.5 | 60.5 | 45.7 | 25.9 | 72.9 | 80.0 | 62.7 |
| CLIC-LAION | 61.7 | 61.0 | 75.9 | 95.0 | 60.0 | 86.7 | 75.6 | 60.1 | 61.1 | 27.9 | 84.5 | 84.0 | 73.7 |
| SAME NOUN | 61.7 | 60.7 | 75.9 | 95.0 | 60.0 | 86.6 | 75.0 | 60.1 | 61.8 | 28.5 | 84.3 | 83.3 | 73.0 |
| LLM SWAPS | 61.8 | 61.1 | 76.1 | 94.7 | 59.8 | 86.7 | 76.7 | 59.9 | 60.7 | 24.0 | 83.3 | 83.9 | 72.1 |

# C  Algorithmic details

---

**Algorithm 1** Training Procedure with Concatenated Images and Hard Negatives

---

1: **function** GENERATEPOSNEG(image pair $(x_i, x_{i+m})$, caption pair $(y_i, y_{i+m})$)
2:      $u_i = \text{RandomConcat}(x_i, x_{i+m})$
3:      Extract the sentences from the captions $y_i^1, y_i^2, ..., y_{i+m}^1, y_{i+m}^2, ...$
4:      `% Positive Generation`
5:      Create $t_i^{p_1} = \text{Concat}(y_i^1, y_{i+m}^1)$    `% Concatenate first sentences`
6:      Create $t_i^{p_2} = \text{Concat}(y_{i+m}^1, y_i^1)$    `% Shuffle order`
7:      Create $t_i^{p_3} = \text{RandomConcat}(y_i^{k_1}, y_{i+m}^{k_2})$ for random $k_1, k_2$    `% Random order of random`
        `sentence pair`
8:      Create $t_i^{p_4} = \text{RandomConcat}(y_i^{k_3}, y_{i+m}^{k_4})$ for random $k_3, k_4$    `% Random order of random`
        `sentence pair`
9:      `% Hard Negative Generation`
10:     Extract linguistic categories for words in $y_i^1$ and $y_{i+m}^1$ using spaCy
11:     **if** common category exist **then**
12:        Select a random common category CAT (e.g. NOUN, ADJ, ...)
13:        Randomly select CAT words $w_i, w_{i+m}$ from $y_i^1, y_{i+m}^1$ respectively
14:     **else**
15:        Randomly select words $w_i, w_{i+m}$ from $y_i^1, y_{i+m}^1$ respectively
16:     **end if**
17:     Swap $w_i$ and $w_{i+m}$ in $t_i$ to create $t_i^n = \text{Swap}(t_i, w_i, w_{i+m})$
18:     **return** $u_i, t_i^{p_1}, t_i^{p_2}, t_i^{p_3}, t_i^{p_4}, t_i^n$
19: **end function**

**Require:** Training images $I$, captions $T$, hyperparameters $\lambda_{\text{Cont}}, \lambda_{\text{S-Neg}}, \lambda_{\text{Uni}}$
20: **for** each iteration **do**
21:     Sample a batch of images and corresponding captions $\{x_i, y_i\}_{i=1}^m$
22:     **if** iteration $\% \; 2 = 0$ **then**
23:        `% Concatenated training images`
24:        **for** each image caption pair $x_i, y_i$ in the batch **do**
25:           $u_i, t_i^{p_1}, t_i^{p_2}, t_i^{p_3}, t_i^{p_4}, t_i^n =$ GeneratePosNeg$((x_i, x_{i+m}), (y_i, y_{i+m}))$
26:        **end for**
27:        `% Compute loss (Contrastive, Hard Negative, Uni Modal)`
28:        $\mathcal{L} = \lambda_{\text{Cont}}\mathcal{L}_{\text{Cont}} + \lambda_{\text{S-Neg}}\mathcal{L}_{\text{S-Neg}} + \lambda_{\text{Uni}}\mathcal{L}_{\text{Uni}}$
29:     **else**
30:        `% Standard CLIP training`
31:        **for** each image caption pair $x_i, y_i$ in the batch **do**
32:           Extract first sentence of caption $y_i^1$
33:        **end for**
34:        Compute standard CLIP contrastive loss
35:     **end if**
36:     Update model parameters
37: **end for**

---

In this section, we present a pseudo-code of our algorithm, as discussed in Section 3 in the main paper. The pseudo-code can be found in Algorithm 1.

# D  Experimental details and discussions

In this section, we give a detailed description of various training and baseline design choices.

## D.1  Further training details and discussions

As stated in the main paper, we train with alternating steps of CLIP loss (Eq. (1)) and the proposed loss (Eq. (7)). This decision is made to help the fine-tuned model retain its performance on downstream tasks, evident from the ablations in Table 4.

**Computational resources.** All the work was carried out on A100 40G GPUs. The training runs are across 4 GPUs. Running CLIC on smaller architectures like ViT-B/32 and ViT-B/16 took less than 30 minutes for the Laion dataset. Datasets such as RedCaps took longer due to high-resolution images. The larger ViT-L/14 models took around 1 hour per dataset. The evaluation time is negligible, in the order of 20 minutes per model for all datasets on a single GPU.

**Training parameters and details.** In all of the experiments, the loss parameters in Eq. (7) are set to $\lambda_{\text{Cont}} = 1/2$, $\lambda_{\text{S-Neg}} = 1/2$ and $\lambda_{\text{Uni}} = 1$. We train for one epoch on our 1M Laion subset and on our 850k PixelProse dataset, while for MS-COCO, we trained for five epochs; ablation on the number of epochs can be found in Table 19. All experiments are conducted at an image resolution of $224 \times 224$. Our effective batch size is $200 \times 4$, and we use a cosine scheduler, where the warm-up phase is $20\%$ of the training time. The learning rate (LR) starts at $1e-7$, peaks at $1e-6$, and arrives at $1e-8$ at the end. We used the standard AdamW [34] optimizer with beta parameters $(0.9, 0.98)$ and $\epsilon$ set to $1e-8$ with a weight decay of 0.1.

**Training with COCO.** Although for most of the experiments, we use our 1M Laion subset or the PixelProse dataset, we also train with MS-COCO to be comparable to the original NegCLIP checkpoint from [56]. Note that due to training on MS-COCO all retrieval numbers on MS-COCO for NegCLIP from [56], and CLIC-COCO are no longer zero-shot. The same is also true to CoN-CLIP, who use COCO images as distractor in their method.

**Our NegCLIP baselines.** To have a baseline for all models considered in this work, we train our own version of NegCLIP (denoted everywhere with $^\dagger$). It is used only when there are no available checkpoints (Table 3, Table 24 and Table 25), and in our ablation study (Table 20). This version of NegCLIP differs from the NegCLIP [56] as follows:

1. We do not use hard negative images that are added to the batch.
2. We use the same categories as in our method, while the NegClip paper employs six tags: two from the coarse-grained POS level ("ADV" and "ADJ"), two from the fine-grained level ("NN" for singular nouns and "NNS" for plural nouns), and two phrase-level categories (verb phrases and noun phrases).
3. The vision encoder is frozen during training.

We use their negative loss calculation (see Eq. (2)), where the hard negatives are part of the standard clip loss.

## D.2 Zero-shot datasets

The 10 zero-shot classification datasets we use are a subset from the `CLIP_benchmark`[2]. Specifically, we use 1k images of each of the following datasets: Country-211 [44], Caltech-101 [15], OxfordPets [40], DTD [4], FGCV Aircrafts [37], StanfordCars [22], Cifar-10,100 [23], Food-101 [2].

## D.3 Dataset from other methods

To improve compositionality, many methods use re-labelled/captioned versions of standard datasets. The details of these recaptions can be found in Table 9. TripletCLIP uses recpationed data from LaCLIP and DAC uses BLIP2 [25] to create better captions. Methods like CoN-CLIP and SVLC-R+L use the original captions and PaLM-2 and BERT [7] to generate hard-negatives for their base datasets.

## D.4 Other methods do not generalize

From the results on SugarCrepe++ and SugarCrepe, one sees that other compositionality enhancing methods like SVLC-R, DAC, etc., perform relatively well on SugarCrepe but fail to generalize to SugarCrepe++. We believe this is due to the way these methods operate. Specifically, most of these methods create hard-negatives aimed at certain attributes such as color, position, or size (for example, modifying P1: "a woman plays a black guitar" to N: "a woman plays a green guitar"). This aligns closely with what benchmarks like ARO, and SugarCrepe evaluate (with SugarCrepe generating more fluent and sensical negatives).

---

[2]https://github.com/LAION-AI/CLIP_benchmark

Table 9: **Recaptioning/negative generation for different methods.** We show the different datasets and the respective re-captioning model used by compositionally enhancing methods.

| Method | Source | Base Dataset | Re-caption Method |
|---|---|---|---|
| NegCLIP | [56] | MS-COCO | - |
| DAC-LLM | [9] | CC3M | BLIP2 [25] |
| DAC-SAM | [9] | CC3M | BLIP2 [25] |
| SVLC-R | [10] | CC3M | - |
| SVLC-R+L | [10] | CC3M | BERT [7] |
| CoN-CLIP | [49] | CC3M | PaLM-2 [1] |
| TripletCLIP | [41] | CC3M, CC12M | LaCLIP [12] |
| CLIC-COCO | Ours | MS-COCO | - |
| CLIC-LAION | Ours | Laion | CogVLM [53] |
| CLIC-CC12M | Ours | CC12M | PixelProse [50] |
| CLIC-RedCaps | Ours | RedCaps | PixelProse [50] |

While such approaches can indeed improve performance on lexical understanding, they fail to capture word replacements that are not in the dictionary and fall outside of a predefined structure. More importantly, we believe these changes do not foster a deeper, more nuanced understanding of compositionality within the model. True compositional reasoning requires distinguishing between examples that differ not only lexically but also syntactically and semantically, for instance, recognizing that N2: "a woman plays a plastic guitar" is negative, whereas P2: "a guitar is being played by a woman" is positive. These cases involve more complex syntactic and semantic variations and move beyond lexical differences.

For SugarCrepe++, in addition to the evaluation in SugarCrepe, an additional lexically different text is introduced, and since these methods do not account for general compositional/structural variations in their hard-negative, they fall short on SugarCrepe++. On the other hand, the random word swapping in CLIC makes our method focus not on the specific attributes but the general composition of different words in the text. This makes our method generalize better and work equally well for both SugarCrepe++ and SugarCrepe.

### D.5 Detailing the LLaVA experiment

As LLaVA uses only the vision encoder from CLIP models trained at a higher image resolution ($336 \times 336$ pixels), we performed an additional fine-tuning of ViT-L/14 with CLIC, this time also unfreezing the vision encoder. We employed the PixelProse dataset and reduced the effective batch size to $40 \times 8$, while keeping all other settings identical to those in Appendix D.1.

Subsequently, we fine-tuned LLaVA-1.5 using Vicuna-7b as the LLM. Initially, only the multi-modal projector was fine-tuned, followed by fine-tuning of both the projector and the LLM. This procedure follows the standard setup from the official LLaVA codebase[3]. To ensure a fair comparison, we conducted these experiments using both the original CLIP and our CLIC vision encoders. Hence, the two resulting models in Table 6 are both fine-tuned by us, making the subsequent comparison fair.

For the evaluation reported in Table 6, we adopt the VQAScore from Lin et al. [30] for both SugarCrepe++ and WinoGround. For standard VLM benchmarks, we select a subset of tasks from the original LLaVA codebase. Specifically, we assess question answering using accuracy on GQA [19] and TextVQA [47], chain-of-thought reasoning with ScienceQA-Images (SQA-I) [35], and perception via MME [54]. For all evaluations we use the default setup from the original LLaVA codebase.

## E  Additional experiments

In this section, we present more detailed results of the experiments in Section 4 in the main paper and provide additional experiments.

---

[3] https://github.com/haotian-liu/LLaVA/tree/main

**Extension of results from the main paper** As we only show averaged ITT and TOT numbers across different sets of SugarCrepe++ and SugarCrepe in the main paper, the full versions for ViT-B/32 models can be found in Table 17 and Table 18 resp. Similarly, for ViT-B/16 (omitted in the main part) and ViT-L/14, detailed results are shown in Table 24 and Table 25, respectively. Detailed compositionality results for the newer CLIP versions in CLIPA, EVA02-CLIP and CLIPS with their respective better versions achieved via CLIC can be found in Table 26. In Table 23, we give dataset wise accuracy of all the large models for the ZS-10 setting. WinoGround results for large model can be found in Table 27. The pre-trained checkpoints used for fine-tuning with CLIC are listed in Table 22.

**Improvements across architectures** In Table 3 in the main part of the paper, we show how CLIC improves compositionality and retrieval for differently pre-trained models. We visualize these improvements in Figure 5. Fine-tuning with CLIC yields improvements on all models for ITT on SugarCrepe++ and SugarCrepe. The improvements in case of CLIP are as high as 7% and as much as 9% for CLIPS.

**Ablating training steps** In Table 19, we show how the proposed CLIC changes with more training steps. Here, we use MS-COCO for training which means 5 epochs corresponds to 2.5 Laion epochs of our dataset. From the table, we see increasing gains in SugarCrepe++ numbers when increasing the number of training step. However, this comes with a marginal degradation in zero-shot classification, which goes down from 63.2% to 62.7% on IMAGENET as we move from 2 to 5 epochs. The same trend holds for our 1M set of Laion, where training for more than 1 epochs led to decay on IMAGENET and ZS-10 numbers with marginal gains on SugarCrepe++. Hence, for our runs in the main part of the paper with Laion (CLIC-LAION) we only train for 1 epoch.

**Ablating freezing of model components** In Table 20, we show how the proposed CLIC (frozen vision encoder) yields, on average, the best compositionality without sacrificing downstream performance. From the table, it is clear that freezing just the text encoder leads to the least amount of improvement on compositional benchmarks. Fine-tuning the whole model works better (in terms of retrieval) but the gains on SugarCrepe++ are smaller than freezing the vision encoder and it also degrades zero-shot classification performance. From these results, we infer that the most improving configuration is freezing the vision encoder, which is the final setting for CLIC. This also highlights that substantial gains can be made by improving the text encoder alone.

**Ablating number of additional positives** In Table 21, we show the impact of adding additional positives for each concatenated image ($p_3$ and $p_4$ from Section 3.1). Notably, increasing the number of positives has opposing effects on the TOT and ITT scores. In addition, after two positives, the effects on ITT in SugarCrepe++ become minimal, so we chose to add only two positives.

**Error bars** To show the resilience to randomness of CLIC, we conduct three runs for two versions of CLIC presented in Table 1 and Table 2 (CLIC-LAION and CLIC-RedCaps). We report the mean and standard deviation in Table 10 and Table 11, respectively. The standard deviation is small, particularly when considering that the results in the paper are rounded to one decimal place.

Table 10: **Small standard deviation across runs.** Results are reported as mean $\pm$ std over three independent runs on compositionality benchmarks with the ViT-B/32 architecture.

| | SugarCrepe++ [11] | | | | WinoGround [52] | | | SugarCrepe [18] | | |
| | Replace | | Swap | | Text | Image | Group | Add | Replace | Swap |
| Method | ITT | TOT | ITT | TOT | Score | Score | Score | ITT | ITT | ITT |
|---|---|---|---|---|---|---|---|---|---|---|
| CLIP | 69.5 | 60.5 | 45.7 | 25.9 | 31.2 | 11.0 | 8.7 | 72.9 | 80.0 | 62.7 |
| CLIC-LAION | 75.34 ± 0.18 | 60.10 ± 0.11 | 61.55 ± 0.38 | 27.86 ± 0.08 | 31.75 ± 0.20 | 11.83 ± 0.31 | 9.33 ± 0.31 | 84.56 ± 0.09 | 83.84 ± 0.14 | 73.79 ± 0.12 |
| CLIC-RedCaps | 76.18 ± 0.16 | 57.82 ± 0.17 | 61.60 ± 0.07 | 23.34 ± 0.19 | 32.00 ± 0.61 | 11.67 ± 0.12 | 10.08 ± 0.24 | 86.27 ± 0.15 | 84.80 ± 0.04 | 72.54 ± 0.04 |

**Long captions retrieval** To evaluate the influence of exploiting superficial artifacts such as potential abrupt topic changes, generated by our concatenation scheme, we compared the retrieval performance on longer captions (up to 200 tokens) between the original model, other compositionality-enhancing methods, and our fine-tuned models (CLIC) on the ViT-B/32 architecture. This setup is adapted

Table 11: **Small standard deviation across runs.** Results are reported as mean $\pm$ std over three independent runs on downstream tasks with the ViT-B/32 architecture.

| Method | Classification | | $I \rightarrow T$ | | $T \rightarrow I$ | |
| | IMNET | ZS10 | COCO | F30K | COCO | F30K |
|---|---|---|---|---|---|---|
| CLIP | 63.3 | 61.4 | 74.1 | 95.1 | 54.6 | 83.5 |
| CLIC-LAION | $61.72 \pm 0.02$ | $60.89 \pm 0.06$ | $75.92 \pm 0.17$ | $95.10 \pm 0.08$ | $60.32 \pm 0.64$ | $86.70 \pm 0.03$ |
| CLIC-RedCaps | $62.41 \pm 0.09$ | $60.22 \pm 0.03$ | $76.00 \pm 0.09$ | $95.67 \pm 0.05$ | $59.42 \pm 0.04$ | $86.27 \pm 0.02$ |

from long-retrieval as previously done in [58]. We truncate the caption to the nearest punctuation (within context-length) as CLIP has a context length of 77. It is important to note that these longer captions describe a single scene but consist of multiple sentences. This design choice was deliberate: it helps us determine if our CLIC models have inadvertently overfit to abrupt topic changes or even to concatenation with punctuation. As can be seen in Table 12, only CLIC and TripletCLIP outperform the base model for both image and text retrieval, whereas DAC-LLM and SVLC-R+L are worse.

Table 12: **CLIC does not overfit to abrupt topic changes.** Evaluation of long-caption retrieval task for the ViT-B/32 architecture.

| Method | $I \rightarrow T$ | $T \rightarrow I$ |
|---|---|---|
| CLIP | 83.2 | 79.5 |
| NegCLIP | 82.6 | 80.7 |
| DAC-LLM | 67.6 | 72.5 |
| SVLC-R+L | 75.0 | 74.8 |
| TripletCLIP | 84.1 | **81.5** |
| CLIC-COCO | **85.9** | 81.4 |
| CLIC-LAION | 83.9 | 79.6 |
| CLIC-RedCaps | 84.7 | 79.9 |

**More retrieval results** In the main part of the paper, all results on MS-COCO and Flickr30k retrieval are reported with Recall@5. In Table 28, we additionally report Recall@1 for all ViT-B/32 models from Table 1. The overall ordering of methods is similar to Recall@5. Similarly, in Table 27, we additionally report Recall@1 for all ViT-L/14 models from Table 3.

**Visual examples and failure case analysis** To illustrate the strengths and limitations of the method and the compositionality task in general, we selected four of the SugarCrepe++ categories: Replace attribute, Replace relation, Swap attribute, and Swap object. For each, we examined the first instance in which the pretrained model's prediction changed from incorrect to correct with CLIC-RedCaps, and vice versa. The results show that CLIC has lower absolute cosine similarity values compared to the pretrained model.

Table 13: Visual examples illustrating cases where the pretrained model's prediction changed: from incorrect to correct with our CLIC-RedCaps version, and from correct to incorrect in the **replace attribute** class of SugarCrepe++. ✓ marks correct predictions and ✗ marks incorrect predictions.

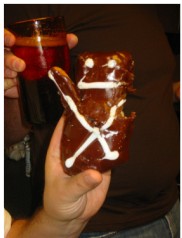

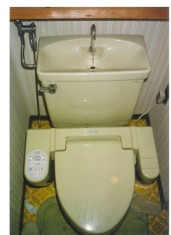

$P_1$: A person holding up a chocolate doughnut with a face drawn on it.
$P_2$: A chocolate doughnut with a face drawn on it is being held up by a person.

$N$: A person holding up a vanilla doughnut with a face drawn on it.

$P_1$: A tan toilet and sink combination in a small room.
$P_2$: A small room with a tan toilet and sink combination positioned in close proximity to one another.
$N$: A white toilet and sink combination in a small room.

| Algorithm | $\cos(P_1)$ | $\cos(P_2)$ | $\cos(N)$ |
|---|---|---|---|
| CLIP (✗) | 0.323 | 0.306 | 0.309 |
| CLIC (✓) | 0.280 | 0.285 | 0.263 |

| Algorithm | $\cos(P_1)$ | $\cos(P_2)$ | $\cos(N)$ |
|---|---|---|---|
| CLIP (✓) | 0.325 | 0.341 | 0.319 |
| CLIC (✗) | 0.260 | 0.256 | 0.262 |

Table 14: Visual examples illustrating cases where the pretrained model's prediction changed: from incorrect to correct with our CLIC-RedCaps version, and from correct to incorrect in the **replace relation** class of SugarCrepe++. ✓ marks correct predictions and ✗ marks incorrect predictions.

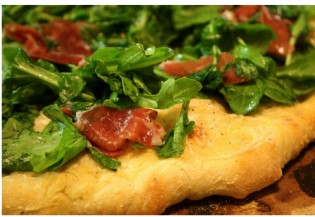

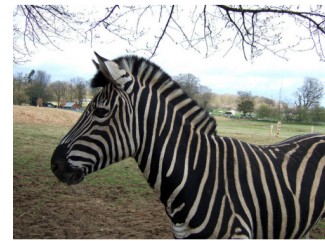

$P_1$: A pizza covered in lots of greens on top of a table.
$P_2$: A table has a pizza on top of it covered in lots of greens.
$N$: A pizza covered in lots of greens next to a table.

$P_1$: A zebra is standing in an open field.

$P_2$: On an open field, stands a zebra.

$N$: A zebra is running across an open field.

| Algorithm | $\cos(P_1)$ | $\cos(P_2)$ | $\cos(N)$ |
|---|---|---|---|
| CLIP (✗) | 0.300 | 0.280 | 0.293 |
| CLIC (✓) | 0.276 | 0.275 | 0.272 |

| Algorithm | $\cos(P_1)$ | $\cos(P_2)$ | $\cos(N)$ |
|---|---|---|---|
| CLIP (✓) | 0.311 | 0.312 | 0.298 |
| CLIC (✗) | 0.283 | 0.254 | 0.270 |

## Laion re-captioned images

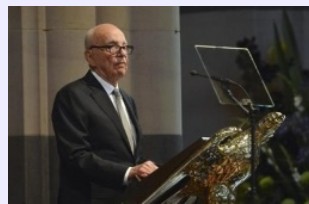

An elderly man wearing glasses and a suit standing behind a lectern with a golden eagle design. He appears to be in a formal setting possibly delivering a speech or presentation. The background is dimly lit and there are some decorative elements such as flowers visible behind him.

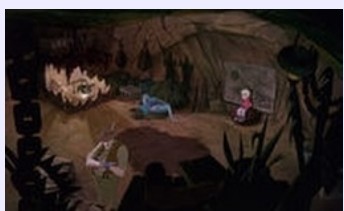

A dimly lit cave or underground setting. The cave has various elements such as stalactites rocks and a chalkboard with some drawings on it. The atmosphere seems mysterious and adventurous suggesting that the characters might be on a quest or exploration.

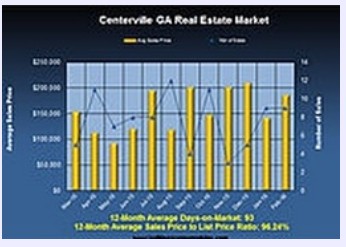

A bar chart that represents the average sales price of homes in Centerville Georgia over a 12-month period from January 2010 to December 2010. The chart shows two sets of data: the average sales price of homes and the average sales price to list price ratio.

## PixelProse images

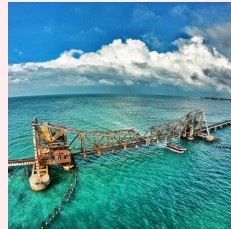

In the foreground, there is a long bridge with a train on it. The bridge is made of metal and has a brown color. The train is red and white. Behind the bridge, there is a large body of water. The water is a blue-green color. There are some clouds in the sky. The image displays the Pamban Bridge, which is a still active train bridge located in Rameshwaram, India.

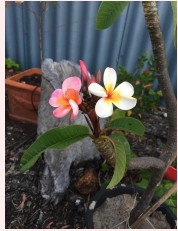

A pink and white frangipani flower. The flower is made up of five petals, with the pink petals on the outside and the white petals on the inside. The flower is surrounded by green leaves.

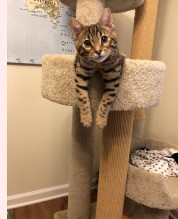

A small Bengal cat with brown fur and black stripes. The cat is lying on a cat tree, with its front paws hanging off the edge of the platform. The cat is looking at the camera with a curious expression.

Figure 4: **Random images from Laion and PixelProse, after re-captioning with CogVLM [53] and re-captions from Singla et al. [50].** For each caption, the first sentence and another two randomly sampled sentences are presented. The first sentence often describes the entire image, while the additional sentences highlight specific details. As can be seen, the captions generated by CogVLM [53] exhibit high quality similar to the PixelProse dataset. Note these models still hallucinate (e.g. train on the bridge) sometimes and not all captions are 100% correct.

Table 15: Visual examples illustrating cases where the pretrained model's prediction changed: from incorrect to correct with our CLIC-RedCaps version, and from correct to incorrect in the **swap attribute** class of SugarCrepe++. ✓ marks correct predictions and ✗ marks incorrect predictions.

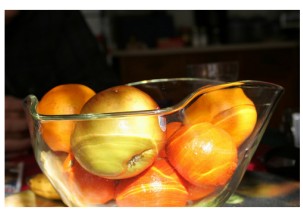

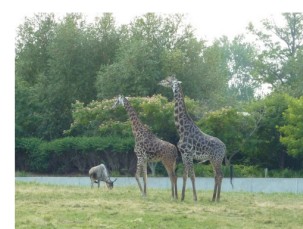

$P_1$: One apple and several oranges sit in a bowl.
$P_2$: several oranges and an apple are positioned in a bowl.
$N$: Several apples and one orange sit in a bowl.

$P_1$: Two giraffe and a zebra are standing in a field.
$P_2$: In a field, two giraffes and a zebra are standing.
$N$: A giraffe and two zebras are standing in a field.

| Algorithm | $\cos(P_1)$ | $\cos(P_2)$ | $\cos(N)$ |
|---|---|---|---|
| CLIP (✗) | 0.306 | 0.315 | 0.309 |
| CLIC (✓) | 0.276 | 0.298 | 0.270 |

| Algorithm | $\cos(P_1)$ | $\cos(P_2)$ | $\cos(N)$ |
|---|---|---|---|
| CLIP (✓) | 0.316 | 0.316 | 0.315 |
| CLIC (✗) | 0.305 | 0.290 | 0.295 |

Table 16: Visual examples illustrating cases where the pretrained model's prediction changed: from incorrect to correct with our CLIC-RedCaps version, and from correct to incorrect in the **swap object** class of SugarCrepe++. ✓ marks correct predictions and ✗ marks incorrect predictions.

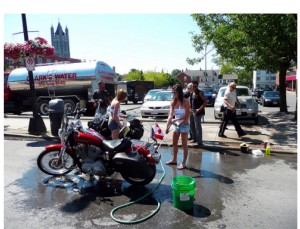

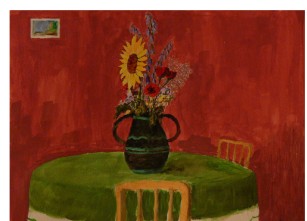

$P_1$: Girls wash a motorcycle while men look on.
$P_2$: The motorcycle is being washed by girls while men observe.
$N$: Men wash a motorcycle while girls look on.

$P_1$: A painting of a vase with a sunflower on a table.
$P_2$: A vase containing a sunflower is positioned on a table in a painting.
$N$: A painting of a sunflower with a vase on a table.

| Algorithm | $\cos(P_1)$ | $\cos(P_2)$ | $\cos(N)$ |
|---|---|---|---|
| CLIP (✗) | 0.365 | 0.356 | 0.358 |
| CLIC (✓) | 0.364 | 0.371 | 0.362 |

| Algorithm | $\cos(P_1)$ | $\cos(P_2)$ | $\cos(N)$ |
|---|---|---|---|
| CLIP (✓) | 0.378 | 0.381 | 0.378 |
| CLIC (✗) | 0.317 | 0.321 | 0.322 |

Table 17: **Extended results for SugarCrepe++ for ViT-B/32 architecture.** This table is an extension of Table 1.

| Method | Replace-obj ITT | TOT | Replace-att ITT | TOT | Replace-rel ITT | TOT | Swap-obj ITT | TOT | Swap-att ITT | TOT | AVG ITT | TOT |
|---|---|---|---|---|---|---|---|---|---|---|---|---|
| CLIP | 86.8 | 83.7 | 65.6 | 59.3 | 56.2 | 38.6 | 46.1 | 19.2 | 45.2 | 32.7 | 60.0 | 46.7 |
| NegCLIP | 89.6 | 94.5 | 69.5 | 76.4 | 52.3 | 51.4 | 54.7 | 33.1 | 58.8 | 56.7 | 64.8 | 62.5 |
| DAC-LLM | 65.7 | 76.8 | 47.7 | 59.5 | 47.6 | 42.3 | 31.4 | 11.4 | 32.9 | 24.8 | 45.1 | 43.0 |
| DAC-SAM | 64.3 | 75.8 | 43.9 | 56.1 | 48.6 | 48.6 | 27.7 | 11.4 | 33.6 | 25.4 | 43.6 | 43.5 |
| SVLC-R | 82.9 | 89.5 | 61.5 | 67.4 | 47.6 | 51.5 | 49.4 | 20.4 | 53.3 | 36.3 | 58.9 | 53.0 |
| SVLC-R+L | 80.9 | 91.6 | 57.1 | 67.0 | 47.3 | 51.3 | 42.8 | 18.4 | 48.9 | 34.4 | 55.4 | 52.5 |
| CoN-CLIP | 88.1 | 91.5 | 66.6 | 69.4 | 52.3 | 51.6 | 44.1 | 22.4 | 51.9 | 42.6 | 60.6 | 55.5 |
| TripletCLIP | 86.8 | 92.3 | 71.7 | 74.0 | 61.9 | 51.7 | 38.8 | 24.1 | 47.9 | 42.3 | 61.5 | 56.9 |
| CLIC-COCO | **90.2** | **97.3** | 72.3 | **81.6** | 58.0 | **59.3** | 48.6 | **33.5** | 57.2 | 60.8 | 65.3 | 66.5 |
| CLIC-LAION | 90.0 | 84.5 | **75.8** | 57.9 | 61.0 | 38.0 | **61.2** | 22.4 | 61.0 | 33.4 | 69.7 | 47.2 |
| CLIC-RedCaps | 90.1 | 84.3 | 75.1 | 52.4 | **62.9** | 36.3 | 60.8 | 17.9 | **62.4** | 28.2 | **70.3** | 34.6 |

Table 18: **Extended results for SugarCrepe for ViT-B/32 architecture.** This table is an extension of Table 1.

| Methods | IMAGENET | Add Obj. | Att. | Replace Obj. | Att. | Rel. | Swap Obj. | Att. |
|---|---|---|---|---|---|---|---|---|
| CLIP | **63.3** | 77.2 | 68.6 | 90.9 | 80.1 | 69.1 | 61.2 | 64.3 |
| NegCLIP | 60.9 | 88.8 | 84.7 | 93.8 | 87.7 | 73.9 | **75.5** | **76.4** |
| DAC-LLM | 51.1 | **89.6** | **97.7** | **94.4** | **89.3** | **84.4** | 75.1 | 74.2 |
| DAC-SAM | 52.3 | 87.5 | 95.5 | 91.2 | 85.9 | 83.9 | 71.8 | 75.4 |
| SVLC-R | 58.8 | 79.4 | 91.2 | 91.3 | 81.2 | 64.2 | 68.6 | 69.1 |
| SVLC-R+L | 59.7 | 75.8 | 81.1 | 88.1 | 76.8 | 62.6 | 64.1 | 66.7 |
| CoN-CLIP | 63.2 | 87.9 | 78.0 | 91.8 | 81.0 | 66.3 | 63.7 | 67.0 |
| TripletCLIP | 54.7 | 87.3 | 85.8 | **94.4** | 86.7 | 82.8 | 66.5 | 72.7 |
| CLIC-COCO | 62.7 | 88.8 | 82.4 | 93.6 | 85.9 | 71.6 | 65.7 | 72.8 |
| CLIC-LAION | 61.7 | 81.6 | 87.4 | 93.4 | 86.3 | 72.4 | 73.1 | 74.2 |
| CLIC-RedCaps | 62.3 | 84.2 | 88.7 | 93.3 | 86.9 | 74.0 | 71.8 | 73.3 |

Table 19: **Witnessing the effect of training steps on CLIC.** Although with more training steps, SugarCrepe++ numbers improve, it comes at the marginal degradation in downstream classification tasks. Hence, we do not train for more steps. $^*$ are not zero-shot for MS-COCO evaluations.

| | | Downstream Evaluations Classification | | Text Ret. | | Image Ret. | | SugarCrepe++ Replace | | Swap | | SugarCrepe Add | Rep. | Swap |
|---|---|---|---|---|---|---|---|---|---|---|---|---|---|---|
| Method | Fine-tuning Epochs | IMNET | ZS-10 | COCO | F30k | COCO | F30k | ITT | TOT | ITT | TOT | ITT | ITT | ITT |
| CLIP [45] | - | 63.3 | 61.4 | 74.1 | 95.1 | 54.6 | 83.5 | 69.5 | 60.5 | 45.7 | 25.9 | 72.9 | 80.0 | 62.7 |
| CLIC-COCO$^*$ | 2 | 63.2 | 60.9 | 81.7 | 96.6 | 66.3 | 89.8 | 73.0 | 78.3 | 53.0 | 45.4 | 85.6 | 83.3 | 69.9 |
| CLIC-COCO$^*$ | 3 | 62.9 | 60.9 | 82.3 | 97.1 | 67.5 | 90.3 | 73.5 | 78.7 | 52.6 | 45.8 | 85.2 | 83.5 | 70.1 |
| CLIC-COCO$^*$ | 5 | 62.7 | 60.7 | 82.9 | 97.1 | 68.2 | 90.2 | 73.5 | 79.4 | 52.9 | 47.2 | 85.6 | 83.7 | 69.3 |

Table 20: **Effect of freezing/unfreezing different model components on CLIC for ViT-B/32.** All these runs are done with the CLIC-LAION version, and overall CLIC with frozen vision encoder works the best.

| Method | Frozen | Classification | | Text Ret. | | Image Ret. | | Repl. | | Swap | | Add | Repl. | Swap |
|---|---|---|---|---|---|---|---|---|---|---|---|---|---|---|
| | | IMNET | ZS-10 | COCO | F30k | COCO | F30k | ITT | TOT | ITT | TOT | ITT | ITT | ITT |
| CLIP [45] | | 63.3 | 61.4 | 74.1 | 95.1 | 54.6 | 83.5 | 69.5 | 60.5 | 45.7 | 25.9 | 72.9 | 80.0 | 62.7 |
| CLIC | None | 60.5 | 60.8 | 77.7 | 94.7 | 62.1 | 87.4 | 72.2 | 60.0 | 60.0 | 25.1 | 84.5 | 83.6 | 71.9 |
| CLIC | Text | 60.2 | 60.3 | 75.8 | 93.4 | 61.1 | 86.5 | 72.5 | 60.5 | 49.8 | 25.9 | 76.8 | 80.1 | 64.4 |
| CLIC | Vision | 61.7 | 61.0 | 75.9 | 95.0 | 60.0 | 86.7 | 75.6 | 60.1 | 61.1 | 27.9 | 84.5 | 84.0 | 73.7 |

The header spanning groups are: Downstream Evaluations (Classification, Text Ret., Image Ret.); SugarCrepe++ (Repl., Swap); SugarCrepe (Add, Repl., Swap).

Table 21: **Effect of adding additional positives on CLIC for ViT-B/32.** All experiments use the CLIC-LAION version, with the negative caption ($n$). The column "Positives" reports addition of more positives ($p_2, p_3, p_4, p_5$) from Section 3.1. Beyond four positives, changes are marginal, so we chose this configuration.

| Method | Positives | Classification | | Text Ret. | | Image Ret. | | Replace | | Swap | | Add | Replace | Swap |
|---|---|---|---|---|---|---|---|---|---|---|---|---|---|---|
| | | IMNET | ZS-10 | COCO | F30k | COCO | F30k | ITT | TOT | ITT | TOT | ITT | ITT | ITT |
| CLIP [45] | | 63.3 | 61.4 | 74.1 | 95.1 | 54.6 | 83.5 | 69.5 | 60.5 | 45.7 | 25.9 | 72.9 | 80.0 | 62.7 |
| CLIC | $p_1$ | 62.0 | 61.3 | 75.5 | 94.4 | 60.0 | 86.4 | 69.0 | 67.1 | 51.8 | 35.8 | 85.9 | 83.1 | 70.9 |
| CLIC | $+p_2$ | 61.8 | 61.2 | 75.5 | 94.6 | 59.9 | 86.5 | 69.7 | 68.6 | 53.3 | 35.3 | 85.6 | 82.8 | 72.3 |
| CLIC | $+p_3$ | 61.7 | 60.8 | 75.8 | 94.9 | 59.8 | 86.6 | 73.8 | 61.7 | 58.7 | 30.2 | 84.5 | 83.8 | 72.6 |
| CLIC | $+p_4$ | 61.7 | 61.0 | 75.9 | 95.0 | 60.0 | 86.7 | 75.6 | 60.1 | 61.1 | 27.9 | 84.5 | 84.0 | 73.7 |
| CLIC | $+p_5$ | 61.7 | 60.7 | 76.2 | 95.1 | 59.8 | 86.7 | 75.6 | 59.3 | 61.5 | 27.0 | 84.1 | 83.6 | 73.0 |

Table 22: **Pre-training/Evaluation checkpoint locations.** Key values for the pre-trained models, taken from different sources.

| Model | Source | Key |
|---|---|---|
| ViT-B/32 | OpenClip | `openai` |
| ViT-B/16 | OpenClip | `openai` |
| ViT-L/14 | OpenClip | `openai` |
| NegCLIP | GitHub | `mertyg/vision-language-models-are-bows` |
| DAC-SAM | GitHub | `SivanDoveh/DAC` |
| SVLC-R+L | GitHub | `SivanDoveh/TSVLC` |
| CoN-CLIP | GitHub | `jaisidhsingh/CoN-CLIP` |
| TripletCLIP | GitHub | `tripletclip/TripletCLIP` |
| EVA-CLIP | OpenClip | `eva02_large_patch14_clip_224` |
| CLIPA | HF-Hub | `UCSC-VLAA / hf-hub:UCSC-VLAA/ViT-L-14-CLIPA-datacomp1B` |
| CLIPS | HF-Hub | `UCSC-VLAA / ViT-L-14-CLIPS-224-Recap-DataComp-1B` |

Table 23: **Zeroshot image classification.** We report the zero-shot image classification performance for different models.

| Model | Source | Food101 | Cifar10 | Cifar100 | Cars | FGVS | DTD | Pets | Caltech101 | Flowers | Country211 | Mean |
|---|---|---|---|---|---|---|---|---|---|---|---|---|
| CLIP | [45] | 93.6 | 96.4 | 76.3 | 76.5 | 30.4 | 54.6 | 94.9 | 86.4 | 80.6 | 33.1 | 72.3 |
| CoN-CLIP | [49] | 93.9 | 96.0 | 79.4 | 76.3 | 30.4 | 56.8 | 93.6 | 86.8 | 78.1 | 33.0 | 72.4 |
| CLIC-LAION | ours | 93.3 | 96.4 | 78.3 | 75.0 | 34.1 | 56.5 | 89.6 | 86.6 | 75.1 | 33.6 | 71.8 |
| CLIC-RedCaps | ours | 92.5 | 96.3 | 78.0 | 76.0 | 30.8 | 56.6 | 91.7 | 87.3 | 78.6 | 32.9 | 72.1 |
| CLIPA | [26] | 94.9 | 98.7 | 88.7 | 92.7 | 42.2 | 69.5 | 95.0 | 88.5 | 81.8 | 31.3 | 78.3 |
| CLIC-LAION | ours | 93.9 | 98.8 | 88.4 | 92.9 | 43.1 | 67.8 | 95.7 | 88.4 | 80.7 | 30.8 | 78.0 |
| CLIC-RedCaps | ours | 94.5 | 98.4 | 88.0 | 93.3 | 41.9 | 69.5 | 95.9 | 89.3 | 80.6 | 31.5 | 78.3 |
| EVA-CLIP | [51] | 94.1 | 99.7 | 90.6 | 89.7 | 37.4 | 61.8 | 94.7 | 88.6 | 77.0 | 32.7 | 76.6 |
| CLIC-LAION | ours | 94.1 | 99.7 | 90.6 | 89.7 | 37.4 | 61.8 | 94.7 | 88.6 | 77.0 | 32.7 | 76.6 |
| CLIC-RedCaps | ours | 92.9 | 99.4 | 89.9 | 88.8 | 33.4 | 62.6 | 92.3 | 89.1 | 77.4 | 32.5 | 75.8 |
| CLIPS | [33] | 93.2 | 98.2 | 86.9 | 91.6 | 39.4 | 66.2 | 94.9 | 86.9 | 78.2 | 29.4 | 76.5 |
| CLIC-LAION | ours | 91.8 | 98.1 | 87.1 | 90.6 | 37.6 | 64.2 | 93.8 | 86.8 | 78.1 | 27.2 | 75.5 |
| CLIC-RedCaps | ours | 92.7 | 98.0 | 86.8 | 91.2 | 37.7 | 67.1 | 94.5 | 87.8 | 80.8 | 30.0 | 76.7 |

Table 24: **Compositionality and downstream results for ViT-B/16.** We present a detailed set of results for the pre-trained CLIP, our version of NegCLIP, CLIP fine-tuned by CoN-CLIP, CLIC-LAION and CLIC-RedCaps (using pre-trained CLIP). For SugarCrepe++, we report only the ITT scores in detail. We note here that models with $^*$ are not zero-shot for MS-COCO evaluations.

| Model | Source | ZS-Class. | | Text-Ret. | | Image-Ret. | |
|---|---|---|---|---|---|---|---|
| | | IMNET | Avg. | COCO | F30k | COCO | F30k |
| CLIP | [45] | 68.3 | 65.5 | 76.2 | 96.4 | 57.6 | 85.6 |
| NegCLIP† | Ours | 64.5 | 64.0 | 74.2 | 94.7 | **65.8** | **89.7** |
| CoN-CLIP* | [49] | **68.9** | **69.1** | 73.9 | 93.5 | 60.1 | 88.1 |
| CLIC-LAION | Ours | 66.5 | 62.3 | 77.6 | 96.9 | 63.6 | 88.9 |
| CLIC-RedCaps | Ours | 67.3 | 64.4 | **79.0** | **97.0** | 62.0 | 88.0 |

| Compositionality: SugarCrepe++ | | | | | | | |
|---|---|---|---|---|---|---|---|
| Model | Source | Rep-obj | Rep-Att | Rep-Rel | Swap-obj | Swap-Att | Avg-ITT | Avg-TOT |
| CLIP | [45] | 89.6 | 67.5 | 53.1 | 39.6 | 48.3 | 59.6 | 45.7 |
| NegCLIP† | Ours | 88.7 | 66.0 | 46.9 | 47.3 | 53.7 | 60.5 | 51.9 |
| CoN-CLIP | [49] | 88.1 | 66.6 | 52.3 | 44.1 | 51.9 | 60.5 | **55.5** |
| CLIC-LAION | Ours | **91.6** | 75.9 | 62.5 | **55.9** | **62.0** | 69.6 | 45.7 |
| CLIC-RedCaps | Ours | 91.5 | **76.0** | **64.5** | **55.9** | 61.1 | **69.8** | 38.6 |

| Compositionality: SugarCrepe | | | | | | | |
|---|---|---|---|---|---|---|---|
| | | Add-obj | Add-Att | Rep-Obj | Rep-Att | Rep-Rel | Swap-Obj | Swap-Att |
| CLIP | [45] | 78.4 | 66.8 | 93.5 | 81.0 | 66.6 | 60.0 | 65.0 |
| NegCLIP† | Ours | 80.0 | 87.3 | 93.9 | 82.2 | 68.1 | 69.0 | 70.1 |
| CoN-CLIP | [49] | 87.3 | 79.6 | 93.6 | 81.0 | 53.3 | 59.2 | 65.2 |
| CLIC-LAION | Ours | **88.4** | **91.0** | **95.6** | **86.5** | **75.5** | **71.0** | **73.9** |
| CLIC-RedCaps | Ours | 87.2 | 88.6 | 94.9 | 85.7 | 73.3 | 69.0 | 71.3 |

Table 25: **Comparison of Sugarcrepe and Sugarcrepe++ along with Downstream Retrieval/-Classification Performance for ViT-L/14.** We present in this table a detailed set of results for the pre-trained CLIP, our version of NegCLIP†, CLIP fine-tuned by CoN-CLIP, CLIC-LAION and CLIC-RedCaps (using pre-trained CLIP). For SugarCrepe++, we report only the ITT scores in detail.

| Compositionality: SugarCrepe++ | | | | | | | | |
|---|---|---|---|---|---|---|---|---|
| Model | Source | Rep-obj | Rep-Att | Rep-Rel | Swap-obj | Swap-Att | Avg-ITT | Avg-TOT |
| CLIP | [45] | 90.6 | 67.5 | 54.0 | 43.6 | 45.6 | 60.3 | 50.3 |
| NegCLIP† | Ours | 90.5 | 68.9 | 52.3 | 46.5 | 54.5 | 62.5 | 52.4 |
| CoN-CLIP | [49] | 92.2 | 68.6 | 55.6 | 44.1 | 47.9 | 61.7 | **54.9** |
| CLIC-LAION | Ours | **94.6** | **79.6** | **62.8** | **59.6** | **60.0** | **71.3** | 46.1 |
| CLIC-RedCaps | Ours | 93.5 | 75.0 | 59.5 | 54.7 | 55.7 | 67.7 | 41.3 |

| Compositionality: SugarCrepe | | | | | | | | |
|---|---|---|---|---|---|---|---|---|
| | | Add-obj | Add-Att | Rep-Obj | Rep-Att | Rep-Rel | Swap-Obj | Swap-Att |
| CLIP | [45] | 78.3 | 71.5 | 94.1 | 79.2 | 65.1 | 60.4 | 62.3 |
| NegCLIP† | Ours | 81.2 | 86.0 | 94.3 | 83.1 | 71.1 | 69.4 | **70.6** |
| CoN-CLIP | [49] | **90.2** | 77.6 | 95.2 | 81.7 | 67.0 | 65.3 | 63.1 |
| CLIC-LAION | Ours | 86.1 | **89.3** | **96.5** | **86.4** | **73.8** | **71.0** | 71.6 |
| CLIC-RedCaps | Ours | 85.8 | 82.8 | 95.4 | 82.9 | 70.0 | 64.5 | 67.7 |

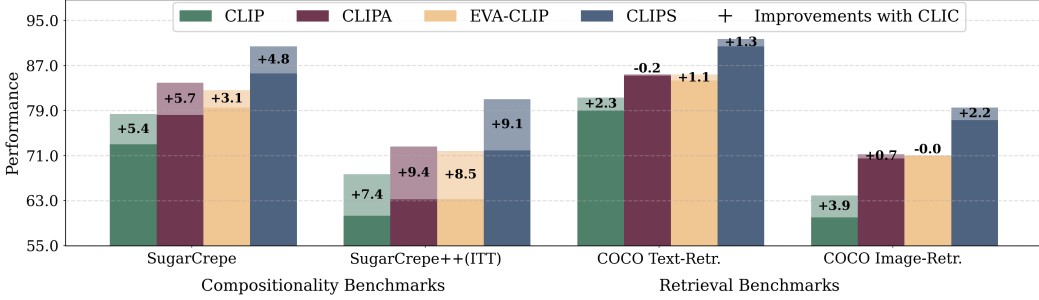

Figure 5: **CLIC improves differently pre-trained CLIP models.** We show for varied CLIP models, using CLIC (PixelProse) gives consistent improvements on both compositionality and downstream evaluation benchmarks. Specifically, to the best of our knowledge, CLIPS +CLIC-RedCaps yields SOTA numbers SugarCrepe++ for CLIP-like models.

Table 26: **Comparison of SugarCrepe and SugarCrepe++ performance for newer CLIP style models.** We present in this table a detailed set of results for the pre-trained CLIPA, EVA-CLIP, CLIPS, and their respective versions fine-tuned with CLIC. For SugarCrepe++, we report only the ITT scores in detail.

| Model | Source | Rep-obj | Rep-Att | Rep-Rel | Swap-obj | Swap-Att | Avg-ITT | Avg-TOT |
|---|---|---|---|---|---|---|---|---|
| Compositionality: SugarCrepe++ | | | | | | | | |
| CLIPA | [26] | 94.5 | 73.6 | 56.3 | 41.2 | 50.4 | 63.2 | 57.2 |
| CLIC-LAION | Ours | 93.9 | 79.1 | 62.1 | 52.2 | 57.9 | 69.1 | 52.1 |
| CLIC-RedCaps | Ours | 95.0 | 80.6 | 64.8 | 60.4 | 62.2 | 72.6 | 46.8 |
| EVA02-CLIP | [51] | 93.2 | 73.5 | 59.3 | 43.7 | 46.7 | 63.3 | 52.7 |
| CLIC-LAION | Ours | 93.8 | 79.7 | 61.8 | 55.9 | 61.1 | 70.4 | 45.2 |
| CLIC-RedCaps | Ours | 95.2 | 80.7 | 63.8 | 59.2 | 60.0 | 71.8 | 41.9 |
| CLIPS | [33] | 95.5 | 77.8 | 64.9 | 51.4 | 69.8 | 71.9 | **65.1** |
| CLIC-LAION | Ours | 95.5 | 84.8 | 72.6 | 66.1 | 80.0 | 79.8 | 62.3 |
| CLIC-RedCaps | Ours | **95.9** | **85.1** | **73.7** | **68.6** | **81.7** | **81.0** | 62.9 |

| Model | Source | Add-obj | Add-Att | Rep-Obj | Rep-Att | Rep-Rel | Swap-Obj | Swap-Att |
|---|---|---|---|---|---|---|---|---|
| Compositionality: SugarCrepe | | | | | | | | |
| CLIPA | [26] | 88.8 | 82.6 | 96.8 | 83.0 | 68.8 | 62.4 | 65.1 |
| CLIC-LAION | Ours | 93.5 | 92.2 | 97.1 | 89.3 | 78.3 | 72.2 | 70.9 |
| CLIC-RedCaps | Ours | 94.5 | 92.3 | 97.0 | 86.9 | 73.8 | 69.8 | 73.3 |
| EVA02-CLIP | [51] | 92.3 | 82.8 | 95.9 | 84.9 | 71.6 | 65.3 | 64.0 |
| CLIC-LAION | Ours | 85.5 | 91.3 | 96.4 | 86.5 | 72.7 | 70.6 | 70.5 |
| CLIC-RedCaps | Ours | 90.7 | 90.6 | 96.7 | 86.7 | 73.2 | 70.2 | 70.3 |
| CLIPS | [33] | 92.5 | 86.3 | 97.4 | 88.6 | 78.1 | 74.3 | 85.0 |
| CLIC-LAION | Ours | **96.6** | **96.8** | **98.7** | **93.0** | **85.5** | **82.0** | **89.3** |
| CLIC-RedCaps | Ours | 95.1 | 95.5 | 97.9 | 92.2 | 83.2 | 81.2 | 87.8 |

Table 27: **Retrieval@1 and WinoGround for ViT-L/14 models.** In addition to retrieval results with Recall@5 in Table 3, we report retrieval scores at Recall@1 and WinoGround results. * are not zero-shot for MS-COCO. The improvements made by CLIC here are consistent with Recall@5.

| Method | Retrieval | | | | WinoGround | | |
|---|---|---|---|---|---|---|---|
| | Text | | Image | | Text | Image | Group |
| | COCO | F30k | COCO | F30k | | | |
| CLIP [45] | 56.0 | 85.1 | 35.2 | 64.7 | 28.7 | 11.0 | 8.5 |
| CoN-CLIP* [49] | 55.8 | 82.5 | 37.8 | 69.3 | 29.5 | 11.0 | 8.0 |
| CLIC-LAION | 56.8 | 85.4 | 40.2 | 70.4 | 30.2 | 10.7 | 8.2 |
| CLIC-RedCaps | 58.3 | 86.9 | 39.0 | 68.8 | 32.2 | 12.2 | 9.0 |
| CLIPA | 64.4 | 90.4 | 46.2 | 73.8 | 32.7 | 8.0 | 7.0 |
| CLIC-LAION | 63.0 | 86.7 | 46.8 | 73.6 | 28.7 | 8.7 | 6.7 |
| CLIC-RedCaps | 63.2 | 89.1 | 46.6 | 73.9 | 32.0 | 11.2 | 7.2 |
| EVA-CLIP | 63.7 | 90.5 | 46.8 | 77.3 | 32.7 | 12.5 | 10.2 |
| CLIC-LAION | 63.6 | 90.5 | 46.5 | 77.4 | 31.7 | 12.7 | 8.7 |
| CLIC-RedCaps | 63.2 | 89.8 | 46.4 | 76.7 | 36.0 | 11.7 | 8.2 |
| CLIPS | 73.6 | 95.7 | 54.2 | 82.6 | 36.2 | 16.0 | 12.5 |
| CLIC-LAION | 71.0 | 95.3 | **56.6** | **85.7** | 37.5 | 16.0 | 13.2 |
| CLIC-RedCaps | **74.6** | **96.3** | **56.6** | 84.3 | **41.7** | **17.5** | **15.2** |

Table 28: **Retrieval@1 for ViT-B/32 models.** In addition to retrieval results with Recall@5 in Table 2, we report retrieval scores at Recall@1. $^*$ are not zero-shot for MS-COCO.

| Method | Retrieval | | | |
| | Text | | Image | |
| | COCO | F30k | COCO | F30k |
|---|---|---|---|---|
| CLIP [45] | 49.5 | 78.5 | 30.1 | 59.1 |
| NegCLIP$^*$ [56] | **60.8** | **85.1** | **45.1** | **70.9** |
| DAC-LLM [9] | 33.0 | 64.9 | 36.7 | 53.2 |
| DAC-SAM [9] | 33.0 | 61.5 | 33.2 | 58.4 |
| SVLC-R [10] | 45.0 | 73.5 | 35.4 | 64.5 |
| SVLC-R+L [10] | 42.8 | 70.4 | 35.6 | 63.7 |
| CoN-CLIP$^*$ [49] | 48.5 | 75.7 | 30.3 | 59.7 |
| TripletCLIP [41] | 46.9 | 73.7 | 32.3 | 60.2 |
| CLIC-COCO$^*$ | 60.0 | 84.7 | 41.1 | 68.3 |
| CLIC-LAION | 52.2 | 79.9 | 35.2 | 62.6 |
| CLIC-RedCaps | 52.5 | 79.4 | 33.8 | 62.8 |

*Prompt: Can you describe this image in detail?*

| Input Image | **LLaVA-1.5-7b (CLIP)** | **LLaVA-1.5-7b (CLIC)** |
|---|---|---|
| 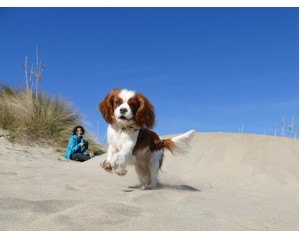 | The image features a small dog with a curly coat, standing on a sandy beach. The dog appears to be enjoying itself as it runs around on the sand. There are two people in the scene, one standing closer to the left side of the dog and the other person further back on the right side. The beach setting is complemented by the presence of a backpack, likely belonging to one of the people, placed on the sand near the center of the scene. | The image features a small brown and white dog running on a sandy beach. The dog appears to be enjoying itself as it runs through the sand, kicking up some dust. There is another person in the background, possibly observing the dog or engaging in their own activities. In the scene, there are also two handbags, one near the person in the background and another closer to the dog. |

*Prompt: Can you describe this image in detail?*

| Input Image | **LLaVA-1.5-7b (CLIP)** | **LLaVA-1.5-7b (CLIC)** |
|---|---|---|
| 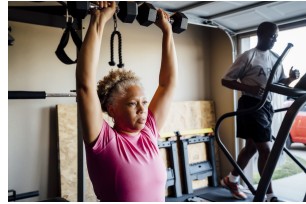 | In the image, there are two people in a gym setting. A woman is standing on a machine, holding a pair of dumbbells, and appears to be working out. A man is standing nearby, also holding a pair of dumbbells. They are both focused on their workout routines. The gym is equipped with various pieces of equipment, including a bench, a treadmill, and a car. The bench is located in the background, while the treadmill is positioned towards the right side of the scene. The car is situated on the far right side of the image, possibly indicating that the gym is located in a parking area. | The image features a woman in a pink shirt and black pants working out in a gym. She is holding a pair of dumbbells and is in the process of lifting them above her head. The woman appears to be focused on her workout, demonstrating her strength and determination. In the background, there is another person who seems to be observing the woman's workout. A car is visible in the far right corner of the image, possibly parked outside the gym. Additionally, there is a bench located near the center of the scene. |

*Prompt: Can you describe this image in detail?*

| Input Image | **LLaVA-1.5-7b (CLIP)** | **LLaVA-1.5-7b (CLIC)** |
|---|---|---|
| 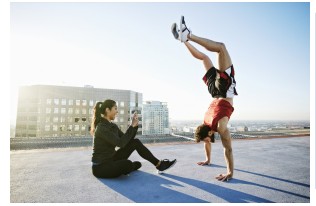 | The image features a man and a woman sitting on a rooftop, both performing handstands. The man is on the left side of the scene, while the woman is on the right side. They are both balancing on their hands, showcasing their impressive strength and flexibility. In the background, there are two cars visible, one on the left side and the other on the right side of the rooftop. The cars are parked at a distance, providing a sense of depth to the scene. | The image features a woman and a man performing a handstand on a rooftop. The woman is sitting on the rooftop, while the man is standing on his hands, doing a handstand. They are both enjoying their time outdoors, showcasing their impressive balance and strength. In the background, there are several cars parked, adding to the urban setting of the scene. The cars are scattered around the rooftop, with some closer to the foreground and others further away. |

*Prompt: Can you describe this image in detail?*

| Input Image | **LLaVA-1.5-7b (CLIP)** | **LLaVA-1.5-7b (CLIC)** |
|---|---|---|
| 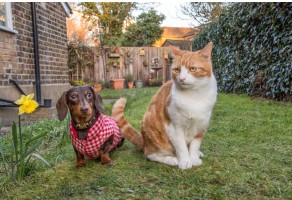 | The image features a small dog wearing a red and white checkered shirt, standing next to a cat. They are both positioned in a grassy yard, with the dog on the left side and the cat on the right side of the scene. In the background, there are two potted plants, one located on the left side and the other on the right side of the yard. A bench can be seen in the middle of the yard, providing a place for people to sit and enjoy the outdoor space. | The image features a small brown and white dog standing next to a large orange and white cat in a yard. Both animals appear to be looking at each other, possibly engaging in a playful interaction. The dog is positioned on the left side of the scene, while the cat is on the right. In the background, there are two potted plants, one located near the left side of the scene and the other on the right side. A bench can also be seen in the yard, situated towards the right side of the scene. |

Figure 6: **Comparing response quality of different LLaVA-1.5-7b versions on random WinoGround validation images.** We compare the open-ended prompt (`Can you describe this image in detail?`) based image captioning of LLaVA with CLIP's vision encoder to that of CLIC. Overall, we believe the responses from CLIC enabled LLaVA to be marginally better than the standard LLaVA, but both models show aspects of hallucinations. Incorrect parts are highlighted and the respective corrections (if available) are also highlighted.

