# OpenReview forum: "Advancing Compositional Awareness in CLIP with Efficient Fine-Tuning"
_NeurIPS.cc/2025/Conference — NeurIPS 2025 poster_

### Official Review · Reviewer_PGUp · 2025-06-08

**Clarity:** 2
**Significance:** 2
**Originality:** 3
**Rating:** 4
**Confidence:** 4

**Summary:**

This paper presents CLIC, a fine-tuning strategy aimed at enhancing compositional reasoning in CLIP-based vision-language models. The method combines multiple images and associated captions during training, with the goal of improving both lexical and semantic compositionality while maintaining or improving downstream retrieval performance. Results on SugarCrepe++ and various retrieval tasks are promising.

**Questions:**

Given the growing prominence of large vision-language models (LVLMs), a discussion—preferably supported by quantitative or qualitative evidence—on whether CLIC-style compositional fine-tuning benefits or transfers to LVLMs (e.g., BLIP-2, LLaVA) would greatly enhance the paper’s relevance to the current landscape.

The paper addresses an important problem and presents a promising direction, but the limited novelty and open questions around generalization and practical utility suggest that further clarification or extensions would strengthen its impact. I'm leaning towards borderline reject before rebuttal and open to discussion with authors and other reviewers.

**Ethical Concerns:**

["NO or VERY MINOR ethics concerns only"]

**Final Justification:**

The rebuttal fully address my concerns, I lean towards accept as promised.

**Limitations:**

yes

**Quality:**

3

**Strengths And Weaknesses:**

Strengths:
* The paper addresses a well-recognized weakness in CLIP models—compositional generalization—and proposes a reasonably efficient solution.
* Experimental evaluation is comprehensive across multiple CLIP variants, including strong baselines like CLIPS.
* The paper is generally well writen and easy to follow.

Weaknesses:
* The core contribution—mixing multiple images and captions during training—can largely be interpreted as a form of compositional data augmentation. While effective, the methodological novelty is relatively incremental, and does not introduce fundamentally new learning principles.
* The inclusion of multiple loss terms introduces several hyperparameters. These are not thoroughly analyzed, and may affect the generalizability or reproducibility of results. A more systematic ablation or sensitivity study would help clarify this.
* Since CLIC is applied as a fine-tuning method on top of pre-trained CLIP models, it is crucial to ensure that their strong generalization and zero-shot capabilities are preserved. The paper emphasizes improvement in compositionality and retrieval but does not clearly analyze potential trade-offs in broader zero-shot tasks.

---

> ### Author Rebuttal · Authors · 2025-07-30
>
> We appreciate the reviewer’s recognition of the importance of the topic, as well as their acknowledgment of our method’s efficiency, the extensive evaluations, and the clarity of the paper. We address the raised questions below.
>
>
> ### Question-1
>
> (Weakness 1) The core contribution .. form of compositional data augmentation... learning principles
>
> Answer-1
>
> Thank you for the new perspective regarding our work. We agree that CLIC can be seen as a better and effective way of compositional data augmentation. While data augmentation is not a new learning principle, it has been instrumental in numerous significant advancements in deep learning. Thus in our point of view suggesting an effective and computationally lightweight form of compositional data augmentation with a thorough evaluation is at least as good as a new ``learning principle’’ . Additionally, our work also highlights a critical issue: how existing methods have overfitted on benchmarks such as SugarCrepe, compromising their generalizability and real-world utility in terms of degraded zero-shot classification or retrieval performance of the resulting CLIP model.
>
> ### Question-2
>
> (Weakness 2) The inclusion of multiple loss terms introduces several hyperparameters. … Systematic ablation
>
> Answer-2
>
> The hyper-parameters used to balance the losses in Equation 7 are:
> $\lambda_{cont}=½$, $\lambda_{S-Neg}=½$, and $\lambda_{uni}=1$, as stated in Appendix D.
> We did not perform a grid search to find the optimal parameters, and it is possible that fine-tuning these hyperparameters (especially with different configurations for each model or dataset) could lead to improved results. However, we used the same hyper-parameters and training settings across all architectures (ViT-B-32, ViT-B-16, ViT-L-14), pretraining datasets and methods (OpenAI, CLIPS, CLIPA, etc.), finetuning datasets (CogVLM, RedCAPS, CC12M, MS-COCO), and different tokenizers (OpenAI, CLIPS). Therefore, we believe that the method is reproducible and generalizable. We would like to highlight that compared to previous work we evaluate on significantly more models and study the dependency on the finetuning dataset.
> In addition, based on the reviewers’ suggestion, we conducted three runs on two of the datasets presented in Tables 1 and 2 (CogVLM and RedCaps), and we report the mean and standard deviation. As can be seen in the table below, the standard deviation is small, particularly when considering that the results in the paper are rounded to one decimal place. We will include these results in the revised version.
>  ### Table E: Small variance across 3 different runs of CLIC finetuning for ViT-B-32 for two datasets CogVLM on LAION and RedCaps
> | Benchmark | Metric | Pre-trained | LAION mean (± std) | RedCaps mean (± std) |
> | --- | --- | --- | --- | --- |
> | **SugarCrepe++** | replace ITT| 69.5| 75.34 ± 0.18 | 76.18 ± 0.16|
> | | replace TOT | 60.5 | 60.10 ± 0.11| 57.82 ± 0.17 |
> | | swap ITT | 45.7 | 61.55 ± 0.38 | 61.60 ± 0.07|
> | | swap TOT | 25.9 | 27.86 ± 0.08 | 23.34 ± 0.19|
> | **SugarCrepe** | add ITT | 72.9| 84.56 ± 0.09| 86.27 ± 0.15 |
> | | replace ITT | 80.0| 83.84 ± 0.14| 84.80 ± 0.04|
> | | swap ITT| 62.7| 73.79 ± 0.12| 72.54 ± 0.04|
> | **Wino** |txt score| 31.2| 31.75 ± 0.20| 32.00 ± 0.61 |
> || img score| 11.0| 11.83 ± 0.31| 11.67 ± 0.12 |
> || group score| 8.7| 9.33 ± 0.31 | 10.08 ± 0.24 |
> | **COCO** | img ret.| 54.6 | 60.32 ± 0.64 | 59.42 ± 0.04 |
> || txt ret.| 74.1 | 75.92 ± 0.17| 76.00 ± 0.09 |
> | **Flickr30K** | img ret. | 83.5| 86.70 ± 0.03 | 86.27 ± 0.02 |
> || txt ret. | 95.1| 95.10 ± 0.08 | 95.67 ± 0.05 |
> | **ImageNet** | acc | 63.3 | 61.72 ± 0.02 | 62.41 ± 0.09 |
> | **ZS10** | acc| 61.4| 60.89 ± 0.06| 60.22 ± 0.03 |
>
>
> ### Question-3
>
> (Weakness 3) Since CLIC is applied as a fine-tuning method …crucial to ensure that their strong generalization and zero-shot capabilities …
>
> Answer-3
>
> We fully agree that enhancing compositionality should not degrade the generalizability and Zero-Shot (ZS) capabilities of CLIP models. The common generalizability and ZS-abilities of CLIP models are measured by ZS-classification and retrieval, both of which have been extensively tested in our work across models. We would like to point out that, unlike CLIC, previous compositionality enhancing methods often suffer from severe degradation in this aspect. For instance, in Tables-2 and 3, we show for models of different sizes that CLIC has a minimal reduction, if any, in zero shot classification (across 11 commonly used datasets) but almost always **improves over the baseline in the standard retrieval benchmarks**. We've also extended individual classification performance for several datasets in Table 15 in the appendix for a more in-depth look. Moreover, in Table-B in response to reviewer m5ip, we show that the performance for long-caption retrieval (significantly longer captions than the standard MS-COCO ones) is also preserved by CLIC fine-tuning. We would be happy to add other zero-shot tasks per the reviewer’s suggestions.
>
> ### Question-4
> (Question 1) Given the growing prominence of large vision-language models (LVLMs), … a discussion supported by quantitative or qualitative evidence for LVLMs..
>
> Answer-4
>
> Given our limited computational resources and time, we couldn't perform full-scale fine-tuning to integrate CLIC models with LVLMs like LLaVA. As we noted in our submission, **CLIC typically fine-tunes only the text-encoder**. However, LLaVA primarily utilizes the CLIP vision encoder, which means CLIC isn't directly compatible **out of the box**. We made sure to highlight this in the limitations section of our paper as well.
> To address this and evaluate LLaVA-1.5 with CLIC, we took a specific approach:
> We fine-tuned CLIC on the Laion CogVLM set, crucially **unfreezing the vision encoder** during this process.
> Then, we replaced the original CLIP vision encoder in LLaVA-1.5 with our CLIC fine-tuned model.
> All evaluations were conducted at a resolution of 224x224. We compared the performance of the original OpenAI CLIP LLaVA-1.5-7B with our CLIC-integrated version in the following tables. In Table F, for general captioning (COCO, Flickr30k), question-answering tasks (VQav2, VizWiz, TExtQA), hallucination (POPE), and Chain-of-thought (ScienceQA-Images), our CLIC-adapted LLaVa performance is very similar to the original LLaVA model (CLIP), even **without further fine-tuning of the vision-language projector**.
>
> Table F: General utility of LLaVA with and without CLIC vision encoders
> |Model  |  VQAv2 | VizWiz | TextQA|  COCO  |   FK30   | POPE | SQA-I
> |---------------|---------|---------|---------|---------|---------|---------|-------|
> **CLIP** |  73.7    |    40.0   |   36.4    | 118.9   |   77.4    |   0.844   | 65.1
> **CLIC** |  73.4    |    39.7   |   36.8    |  118.1   |   76.3   |  0.848   | 65.3
>
> We further adapted the VQAScore from [1] to evaluate LLaVA on compositionality benchmarks, including WinoGround and SC++. In these evaluations, our CLIC-enabled LLaVA achieved **higher compositionality** compared to the original OpenAI-CLIP LLaVA. It's important to reiterate that this improvement was achieved simply by **replacing the vision encoder in LLaVA with our CLIC fine-tuned model**. We are confident that fine-tuning the entire LLaVA model with the CLIC encoder would yield even greater improvements. However, due to the limitations in computing resources and time, this full fine-tuning is not feasible for us at present. We plan to include a detailed discussion regarding this in the final version of our paper.
>
> Table G: WinoGround evaluation of LLaVA
> | Model         | Text    | Image   | Group   |
> |---------------|---------|---------|---------|
> | **CLIP**     | 31.00%  | 30.00%  | 18.00%  |
> | **CLIC**| 33.00%  | 32.25%  | 18.00%  |
>
> Table H: SugarCrepe++ evaluation of LLaVA
> | Model          | Rep_obj | Rep_att | Rep_rel | Swap_obj | Swap_att |
> |----------------|---------|---------|---------|----------|-----------|
> | **CLIP**      | 80.23%  | 63.96%  | 57.47%  | 47.76%   | 55.11%    |
> | **CLIC** | 79.90%  | 65.74%  | 58.25%  | 45.71%   | 56.16%    |
>
> [1] Zhiqui Lin et al., Evaluating Text-to-Visual Generation with Image-to-Text Generation, ECCV 2024

---

> > ### Comment · Reviewer_PGUp · 2025-08-01
> >
> > Thank you for your new results and discussions. My concerns have been partially resolved. I tend to raise my score to 4 (boarderline accept) before discussion with other reviewers.

---

> > > ### Author Response · Authors · 2025-08-05
> > >
> > > Thank you for reading our response and the quick answer. We are happy to answer further questions and appreciate your intention to raise the score.

---

### Official Review · Reviewer_uTvH · 2025-06-26

**Clarity:** 2
**Significance:** 2
**Originality:** 2
**Rating:** 4
**Confidence:** 3

**Summary:**

This paper introduces CLIC: a finetuning method designed to enhance the compositional understanding of vision-language models like CLIP. The core idea is to train on concatenated image pairs and their corresponding captions. The authors generate positive examples with diverse sentence structures and hard negatives by swapping words between captions. The authors demonstrate that CLIC improves performance on compositionality benchmarks like SugarCrepe++ and WinoGround, while maintaining or improving performance on standard retrieval and classification tasks. Experiments are run across different CLIP architectures and pre-training datasets. The method is efficient by fine-tuning only the text encoder.

**Questions:**

See weaknesses section

**Ethical Concerns:**

["NO or VERY MINOR ethics concerns only"]

**Final Justification:**

Thanks for the rebuttal. I keep my initial score (4) and look forward to see more visual examples and failure case analysis in the final version of the manuscript.

**Limitations:**

yes

**Paper Formatting Concerns:**

no concerns

**Quality:**

2

**Strengths And Weaknesses:**

Strengths:
- Addresses the known limitation of vision-language models regarding compositional reasoning.
- Proposes a novel and efficient data generation and fine-tuning strategy using image/caption concatenation and word swapping for hard negatives.
- Demonstrates effectiveness across various CLIP architectures and pre-training datasets.

Weaknesses:
- The reliance on SpaCy for hard negative generation, while efficient, might be less nuanced and powerful compared to LLM-based approaches.
- Would be nice to add visual examples and failure case analysis

---

> ### Author Rebuttal · Authors · 2025-07-30
>
> We appreciate the reviewer’s recognition of the importance of the topic, as well as their acknowledgment of our method’s strong performance, efficiency, and ability to generalize across different architectures and datasets. We address the raised questions below.
>
> ### Question-1
>
> (Weakness 1) The reliance on SpaCy for hard negative generation, while efficient, might be less nuanced and powerful compared to LLM-based approaches.
>
> Answer-1
>
> We agree that LLMs have stronger abilities than the lightweight version of spaCy used in our work for **computational efficiency**. However, as demonstrated by the experiments in the paper, CLIC not only offers greater computational efficiency but also outperforms methods that utilize LLMs, such as TripletCLIP and DAC-LLM. Furthermore, the core approach of CLIC, which combines pairs of images with their corresponding captions, can be extended using LLMs, albeit with increased computational cost.
> An LLM extension involving more challenging image concatenations and word replacements was explored in an experiment addressing Q1 from reviewer m5ip, yielding similar results but at a significantly higher computational cost. Another potential extension of generating negatives from scratch via LLM queries is too computationally demanding, given the resources available to us with current SOTA LLMs.
> ### Question-2
>
>
> (Weakness 2) Would be nice to add visual examples and failure case analysis
>
> Answer-2
>
> Thank you for the suggestion. We agree that it will help clarify the strengths and limitations of the method and the compositionality task in general, and we will incorporate this into the final version of our paper. Unfortunately, due to the current rebuttal format, we cannot upload these results.

---

> > ### Author Response · Authors · 2025-08-08
> >
> > Thanks a lot for your time. As we have addressed all concerns, we would like to ask if the reviewer could  reconsider their score. We are happy to answer further questions in the remaining time.

---

### Official Review · Reviewer_2CLt · 2025-07-03

**Clarity:** 3
**Significance:** 2
**Originality:** 2
**Rating:** 4
**Confidence:** 4

**Summary:**

This paper proposes CLIC, a method to train and improve CLIP models' compositionality. The key idea is to randomly concatenate two images, and utilize their associated captions to make alternative positive captions and hard negative captions to train CLIP models to be able to distinguish lexically similar but semantically different captions, while at the same time being able to understand lexically different but semantically similar captions. Experiment results show some improvements on certain datasets, with more improvements on the text encoder.

**Questions:**

Please see the questions in weaknesses above.

**Ethical Concerns:**

["NO or VERY MINOR ethics concerns only"]

**Final Justification:**

The author responses clarify my concerns on the methodology similarity to existing baselines. Additional experiments with models trained on LAION make the results more convincing without direct train/test distribution match. I am raising my score to 4 after the author rebuttal.

**Limitations:**

Yes.

**Paper Formatting Concerns:**

No formatting concerns.

**Quality:**

3

**Strengths And Weaknesses:**

Strengths:
- The paper tackles an important problem on improving CLIP models' compositional understanding capability as recent work has pointed out that CLIP models are not sensitive to finer-grained semantic understanding.

- The proposed method does not require much additional cost in generating hard positive and hard negative captions, whereas previous work may rely on external models in rewriting captions and generating hard negative captions (or images).

- Experiment results show some improvements on the recently introduced SugarCrepe++ dataset (with more gain shown on text-to-text setting). The resultant CLIC trained model also mostly retains standard classification and retrieval accuracy.


Weaknesses:
- The introduced method that concatenates two randomly sampled images appears somewhat adhoc to me. This changes the image aspect ratio and may negatively affect the model's capability on standard images. This can be seen from CLIC models degraded performances on retrieval/classification performance in Table 2, as compared to original CLIP and baselines like NegCLIP (comparing both MS-COCO trained models).

- The hard negative captions generated by CLIC through swapping words from two captions can introduce unrealistic captions as seen in Figure 2 ("A water wearing a black hat"). As pointed out in SugarCrepe work, these implausible captions might introduce biases into the resultant model. Thus, on the SugarCrepe dataset where the dataset specifically removed implausible negative captions, CLIC models' performances appear to be generally worse than other baselines. Could the authors discuss further on this issue?

- The hard negatives generated from CLIC are almost like replacing words in a caption with other random words (in CLIC, it is done by sampling from another randomly selected caption). Mechanically, this is in fact not too different from existing methods in generating hard negatives by sampling the replacing words from some dictionary.


- While the experiment results show some improvements on SugarCrepe++ dataset, I am concerned that for ITT setting, the improvements mostly comes from training on larger dataset (as CLIC-COCO shows mixed results compared to NegCLIP which is also trained on COCO). On TOT settings, while there are improvements, this might mainly come from the introduction of uni-modal loss, which I do appreciate but is not a significant contribution.

---

> ### Author Rebuttal · Authors · 2025-07-30
>
> We thank the reviewer for their time and valuable comments. We also thank the reviewer for their appreciation of the importance of the topic, and the efficiency+efficacy of the proposed method, CLIC. Below, we address your concerns.
>
> ### Question-1
>
>
> (Weakness1) The introduced method that concatenates two randomly sampled images … may negatively affect the model's capability on standard images. This can be seen from CLIC models …, as compared to original CLIP and baselines like NegCLIP
>
>
> Answer-1
>
> As stated in the paper (lines 127-129), we believe that **there is a problem in assessing methods that were trained on COCO**, since many current evaluations are based on COCO images and captions (SugarCrepe, SugarCrepe++, COCO retrieval, etc.). Thus, training on COCO can leak test information to the model. We trained on COCO to be directly comparable to NegCLIP, but do not consider this as a proper setting, and thus train all other CLIC models on datasets independent of COCO. As can be seen in the ablation results in Table 10 (the relevant parts also included here for convenience), when training NegCLIP and our CLIC on Laion recaptioned with CogVLM, which is independent of  MS-COCO, CLIC performs better on compositionality ITT as well as downstream tasks. Both methods are then evaluated in a zero-shot setting across compositionality benchmarks as well as on downstream tasks. In addition, as can be seen in Tables 1-3 and other experiments in the appendix, when comparing CLIC with other methods that were not trained on COCO as well as a variant of NegCLIP trained on the same dataset as CLIC, **our method achieves the best SugarCrepe++ results as well as Text-retrieval results**, even though all these benchmarks are using a single image input. So we don’t think there is a problem with the model's capability on standard images.
>
> ### Table D: Comparing CLIC to the single image baseline and NegCLIP† when all are trained on Laion recaptioned with CogVLM
>
> | ID  | Method              | IMNET | ZS-10 | COCO Text Ret. | COCO Image Ret. | SC++ Rep. ITT | SC++ Rep. TOT | SC++ Swap ITT | SC++ Swap TOT | SC Add ITT | SC Rep. ITT | SC Swap ITT |
> |-----|---------------------|-------|--------|----------------|-----------------|----------------|----------------|----------------|----------------|--------------|----------------|----------------|
> | A1  | CLIP                | 63.3  | 61.4   | 74.1           | 54.6            | 69.5           | 60.5           | 45.7           | 25.9           | 72.9         | 80.0           | 62.7           |
> |     | **NegCLIP†**        | 61.0  | 60.5   | 72.0           | 59.7            | 67.9           | 64.7           | 54.5           | 36.1           | 82.4         | 80.9           | 70.7           |
> | B4  | Single-Img Baseline | 61.3  | 60.6   | 67.3           | 58.0            | 74.1           | 53.9           | 60.2           | 27.8           | 81.0         | 79.9           | 69.7           |
> | C5  | **CLIC**            | 61.7  | 61.0   | 75.9           | 60.0            | 75.6           | 60.1           | 61.1           | 27.9           | 84.5         | 84.0           | 73.7           |
>
> ### Question-2
>
>
>
> (Weakness 3) The hard negatives generated from CLIC are almost …  Mechanically, this is in fact not too different … replacing words from some dictionary.
>
> Answer-2
>
>
>
> We politely disagree with this perspective. When you rely on dictionary-based word replacements, the modifications are inherently confined to specific, pre-defined changes. For example: altering P1: "a woman plays a black guitar" to N: "a woman plays a green guitar." While this approach can certainly be beneficial for improving performance on lexical understanding (and for existing benchmarks), it doesn't address word replacements that aren't in the dictionary – and perhaps, by extension, aren't in current benchmarks either. More importantly, we believe it doesn't foster a deeper, more nuanced understanding of compositionality within the model. In contrast, true compositional reasoning requires additionally distinguishing between examples like  N2: “a woman plays a plastic guitar” (negative) and P2: “a guitar is being played by a woman” (positive), which involve more complex syntactic and semantic variations. Dictionary-based swaps, though potentially useful for lexical understanding and current benchmark performance, do little to address these challenges, whereas CLIC’s approach, with its variety of negatives that increase quadratically with the size of the dataset (and their varied difficulties starting from non-plausible or grammatically incorrect sentences to hard non-predefined negatives), does. Additionally, at its core, CLIC always swaps two entities that are both present in the scene, whereas dictionary-based methods replace an entity in the scene with one that does not appear in it.
>
> ### Question-3
>
>
> (Weakness 2) The hard negative captions generated by CLIC … Could the authors discuss further on this issue?
>
> (Weakness 4) While the experiment results show some improvements on SugarCrepe++ dataset,…, which I do appreciate but is not a significant contribution.
>
> Answer-3
>
> We agree with the reviewer’s concern that using unrealistic or non-grammatical hard negatives might provide weaker training signals compared to using only grammatical and realistic negatives. However, it is important to note that the Image-to-Text (ITT) metric in SugarCrepe++ uses the **same images, original positive captions, and plausible negative captions from SugarCrepe and adds to them an additional positive caption (lexically different, but semantically similar), as explained in lines 117-122.** We believe our slightly worse performance on SugarCrepe, but better results on SugarCrepe++, are due to the greater variety of negatives and positives seen by our method. Since SugarCrepe++ includes a second positive caption that is lexically more different than both the first positive and the negative captions, to succeed in this benchmark, methods can not rely on shortcuts such as targeting specific attribute replacements that improve results on SugarCrepe but do not generalize to SugarCrepe++. In fact, as can be seen in Table 1, previous methods such as DAC improve over the baseline by 10.4% on Sugarcrepe, but perform worse by -14.9% on Sugarcrepe++ (based on average ITT scores for the replace and swap tasks). This is detailed in Appendix D.4 and further discussed in our response to the previous question regarding dictionary-based methods.
> Our method outperforms previous methods on SC++, suggesting it is **better compositionally aligned than all previous methods, despite being computationally more efficient**, even though all the samples in the benchmark are still grammatically correct and plausible.
>
> **Regarding potential biases introduced by implausible or non-grammatical negatives**
>
> SugarCrepe and SugarCrepe++ are benchmarks, so it is essential that they use plausible and grammatically correct negatives to avoid shortcuts and ensure reliable evaluation. In contrast, since we use negatives only during training, **mixing plausible and implausible (or ungrammatical) negatives should not introduce biases, since all are treated as incorrect** and the model is trained to treat them as incorrect. To avoid biases, our main concern was to accidentally generate negatives that still accurately describe the scene. However, as discussed in Appendix B.3, **swapping words between captions of different images rarely results in captions that accurately describe the concatenated image** as opposed to swapping words within the caption of a single image. Thus, swapping random words should not introduce biases.
> To test whether more plausible and grammatically correct negatives improve performance (at higher computational cost), we conducted an additional experiment that considerably reduced the number of non-plausible and non-grammatical negatives. The results were largely similar, as explained in the first response to reviewer m5ip (associated results in Table-A there).

---

> > ### Comment · Reviewer_2CLt · 2025-08-04
> >
> > Thanks to the authors for the response. It has addressed some of my questions. I have some follow ups:
> >
> > - Q2: The current response doesn't really address my concern here. I do agree that having both hard negatives and hard positives is better than just having either one. However, my main concern is the originality and effectiveness of the specific "swapping" word mechanism proposed in CLIC, which to me is sampling the swapping words from a dictionary defined by the vocabularies  that appear in the dataset captions. Thus, I don't see a big difference to prior hard negative generation methods where you can think of that the hard negatives are generated by selecting and swapping words from captions in the same training batch.

---

### Official Review · Reviewer_m5ip · 2025-07-07

**Clarity:** 4
**Significance:** 2
**Originality:** 2
**Rating:** 5
**Confidence:** 5

**Summary:**

The paper looks at how VLMs like CLIP struggle with with how images are composed, often confusing semantic concepts and their physical relation to each other. This is a well-documented and measured by several benchmark datasets. The authors propose a new fine-tuning method called CLIC, which trains the model using a variety of altered positive captions and negative captions for image-caption pairs. They show this method of training with negative captions helps models with semantic understanding, not just lexical. The SugarCrepe++ benchmark was used.

**Questions:**

Could the authors explain why they are not concerned about the negative captions being ungrammatical or with awkward syntax?
If they can show they get the same results with grammatical and proper sentences, then my issue of the model learning artifacts has been weakened.

Similarly, can the authors provide any analysis that rules out the possibility that the model is learning to exploit superficial artifacts like lists or abrupt topic changes?
My confidence in the model’s compositional abilities would increase.

Do the authors have suggestions for how their approach could be extended to encourage learning of more integrated and hierarchical compositionality?
Concrete proposals or pilot experiments in this direction would help demonstrate that the approach is not limited by its current synthetic design, and would positively impact my assessment.

**Ethical Concerns:**

["NO or VERY MINOR ethics concerns only"]

**Final Justification:**

I responded to the authors' rebuttal by saying they have some good ideas for how to address the concerns, and it would be good to see how they turn out. My original score remains unchanged.

**Limitations:**

Yes

**Quality:**

3

**Strengths And Weaknesses:**

Strengths

The method of combining random images and negative captions by substituting one word offers a cheap way to improve how VLMs understand the meaning of captions. It is flexiblee nough to be used across different types of data.

By joining together two random images and their captions, the training data becomes more complex, which pushes the model to handle combinations of concepts instead of just single objects.

The model achieves high scores on tough benchmarks like Sugarcrepe++, and on some tasks like downstream classification and retrieval, it matches or comes close to the best models available.

Weaknesses

Substituting a single word to make a negative caption may result in unnatural sentences and since the process might often produce sentences that are not meaningful or not grammatical, the model may learn to spot textual artifacts rather than actual compositional relationships.

Also, the model might learn to ficus on lists of captions, separated by a period and with abrupt topic change.

Real photos often contain multiple objects and layers of relationships that are more complex than just combining two random scenes. This might be addressed in a followup study.

SugarCrepe++ is the main test for compositionality in this paper, and while the model does well on it, this benchmark doesn’t require understanding of complex, hierarchical relationships. So, there may still be important gaps in real-world performance.

The paper does not report error bars or statistical significance, so it’s hard to know if the improvements are always reliable, especially when differences are small. The authors address this by saying how it is computationally expensive. However, the differences in Table 1 and 4 are large enough to heurisitally assume there;s likely to be statistical significance. And in Table 2, the differences are small enough to assume statistical insignificance, or at least practical insignificance.

---

> ### Author Rebuttal · Authors · 2025-07-30
>
> We appreciate the reviewer's constructive comments and their recognition of our method’s strong performance on challenging benchmarks, as well as its computational efficiency. We address the raised questions/weaknesses in order below.
>
> ### Question-1
>
> (Weakness 1) Substituting a single word  …  not grammatical …
>
> (Question 1) Could the authors explain  ... negative captions being ungrammatical...
>
> Answer-1
>
> Thanks for the suggestion, we agree that using only meaningful and grammatically correct negatives could further improve the compositionality of the model. To test the impact of non-meaningful or ungrammatical negatives, we made the following modification to our method to get grammatically correct negatives:
> 1) We extracted the fine-grained Part-of-Speech (POS) tags for each word in the first sentence using spaCy’s most computationally expensive model, which took around 5 hours instead of a couple of minutes (in the paper, we prioritized computational efficiency instead).
> 2) For each word, we queried Gemini 2.5-flash to generate up to five replacement candidates from the same POS category, such that replacing the word changes the meaning of the caption, while keeping the sentence coherent. **Note**: these queries to Gemini via API calls took over 24 hours of compute time to find word replacements for the 1M Laion subset. In comparison, fine-tuning a CLIP model with CLIC only takes around 40 minutes.
> 3) During training, for each caption, we randomly selected a POS category and then randomly chose a word from that category. The second image and word replacement were selected based on the candidates from step 2.
> ### Table A: Meaningful and grammatical negatives. All methods use LAION data captioned by CogVLM and the ViT-B-32 architecture.
> | Method | Sampling | Epochs | IMNET | COCO Ret. I→T | COCO Ret. T→I | SC++ Replace (ITT) | SC++ Replace (TOT) | SC++ Swap (ITT) | SC+ Swap (TOT) | Wino Text Score | Wino Image Score| Wino Group Score| SC Add (ITT) | SC Replace (ITT) | SC Swap (ITT) |
> | --- | --- | --- | --- | --- | --- | --- | --- | --- | --- | --- | --- | --- | --- | --- | --- |
> | CLIP | - | - | 63.3 | 74.1 | 54.6 | 69.5 | 60.5 | 45.7 | 25.9 | 31.2 | 11.0 | 8.7 | 72.9 | 80.0 | 62.7|
> | CLIC | Paper| 1| 61.7 | 75.8| 59.8| 75.3| 60.2| 61.9 |28.0| 31.8| 12.3| 9.5| 84.2| 83.8 | 74.0|
> | CLIC | NR |1| 61.8| 76.1| 59.8| 76.7| 59.9| 60.7| 24.0| 31.3| 10.0|7.8 | 83.3| 83.9 |72.1|
>
> This significantly more expensive variant of CLIC (NR for Non-Random in Table A) results in harder negatives (as evidenced by a higher loss during training) and leads to improvements in Replace(ITT) for SugarCrepe++ but degrades Swap(ITT), TOT, and Winoground. While at first surprising, we think that the smaller diversity of negatives compared to our original computationally cheap version of CLIC, with random replacement of words from the same POS category, could be the reason.
>
> To check the quality of the negatives, we labeled 75 negative samples from both NR and our version of the paper. This process yielded 69% meaningful, grammatically correct negatives with distinct semantic meanings from **NR**, compared to 28% from the **Paper version**. Due to character limits, we include below 2 coherent and grammatically correct (_italicized_) hard negatives examples, and 2 nonsensical or grammatically incorrect ones. We will give additional examples in the revised version. While there are some errors, the overall quality is sufficient to prevent the model from relying solely on shortcuts, as many examples remain meaningful and grammatically correct, and the errors are relatively subtle.
>
> (postcard→ telegram)
> _N: a collage of various telegram marketing materials. a vintage postcard._
>
> (stripes→ swirls)
> _N: a black satin jacket with white swirls on the collar and cuffs. a graphic design element that features a large red heart at the center surrounded by intricate golden stripes and patterns._
>
> (whistle→ shout)
> N: a red kettle with a shout attached to its spout. a man in a dramatic pose seemingly in the middle of a punch or a whistle.
>
> (made→ destroyed)
> N: a hand-destroyed card with a floral design. a scene of destruction with a building that appears to have been damaged or made.
>
>
> ### Question-2
>
> (Weakness 2) The model... lists captions … abrupt topic change.
>
> (Question 2) Similarly, can the authors provide any analysis….
>
> Answer-2
>
> To evaluate the influence of exploiting superficial artifacts such as abrupt topic changes, we compared the retrieval performance on larger captions (up to 200 tokens) between the original model, the other compositionality-enhancing methods, and our finetuned models (CLIC) on the ViT-B-32 architecture. This setup is adapted from long-retrieval as done in [1]. We truncate the caption to the nearest punctuation (within context-length) as CLIP has a context length of 77. It's important to note that these longer captions describe a single scene but consist of multiple sentences. This design choice was deliberate: it helps us determine if our CLIC models have inadvertently overfit to abrupt topic changes or even to concatenation with punctuation. Only CLIC and Triplet-CLIP outperform the base model for both image and text retrieval, whereas DAC-LLM and SVLC-R+L are worse.
> ### Table B: Evaluation of long-caption retrieval task
> | Eval | clip | Negclip | DAC-LLM | SVLC-R+L | Triplet | clic-coco | clic-pixpr | clic-cvlm |
> | --- | --- | --- | --- | --- | --- | --- | --- | --- |
> | **I2T** | 0.832 | 0.826 | 0.676 | 0.750| 0.841| 0.859 | 0.847 | 0.839 |
> | **T2I** | 0.795 | 0.807| 0.725 | 0.748 | 0.815 | 0.814 | 0.799 | 0.796 |
>
> We think that this indicates that there is no problem, but it is easier to show that something works well rather than that there is no potential problem. Please let us know if you have other suggestions on how to test this.
>
> ### Question-3
>
> (Weakness 3) Real photos often contain multiple objects and layers of relationships …
>
> (Question 3) Suggestions…hierarchical compositionality? Proposals or pilot experiments …
>
> Answer-3
>
> We think it is possible, at least to a certain extent, with a **variant of CLIC** as we outline below in our proof-of-concept: In the paper, we use a random image for concatenation and a random word for swapping. However, similar to the experiments in App. B.4 and the answer to Q1 above, we can instead pair each image with another that contains the same main object (e.g., a cat) but differs in attributes (e.g., resting vs. in motion). This approach combines the two earlier non-random strategies: (1) keeping the noun consistent, and (2) selecting attributes with different meanings. We can then construct modified captions by changing the second reference to the object, while either preserving (Positive) or changing its attributes (Negative). The following examples highlight how our approach can be extended to such scenarios. However, this modification is more challenging, as the combinations are more prone to false positives and require additional preprocessing. E.g., in image A, the cat may indeed have a curled tail; thus, N2 might be an accurate description of the scene.
>
> An example from Laion-CogVLM dataset (when dropping any extra sentences with non-relevant information):
>
> Image A: "The cat has a tabby pattern with a mix of brown, black, and white fur, and appears to be resting or lying down."
>
> Image B: "The cat appears to be in motion with its tail curled."
>
> We concatenate the images to generate four variants (two positives and two negatives). Unlike the approach in the paper, we now use a single sentence that describes only one of the two cats in the scene. As a result, the model must correctly associate the described action with either the cat that has a tabby pattern or the one with a curled tail. P1 and P2 as above (Image A and Image B).
>
> N1:"The cat has a tabby pattern with a mix of brown, black, and white fur, and appears to be in motion."
>
> N2:"The cat appears to be resting with its tail curled."
>
> Another option is to use different objects and attributes (presenting only the negatives):
>
> Image C: A fluffy white dog possibly a Samoyed walking on a road.
>
> N1:The cat has a tabby pattern with a mix of brown black and white fur, walking on a road.
>
> N2:A fluffy white dog possibly a Samoyed, appears to be resting or lying down.
>
> ### Question-4
>
> (Weakness 4) Report error bars or statistical significance…
>
> Answer-4
>
> We do three runs on two of the datasets presented in Tab. 1 and 2 (CogVLM and RedCaps), and we report the mean and standard deviation in Table C. The standard deviation is small, particularly when considering that the results in the paper are rounded to one decimal place. We include these results in the revised version.
> ### Table C: Small variance across 3 different runs of CLIC finetuning for ViT-B-32 for two datasets CogVLM on LAION and RedCaps
> | Benchmark | Metric | Pre-trained | LAION mean (± std) | RedCaps mean (± std) |
> | --- | --- | --- | --- | --- |
> | **SugarCrepe++** | replace ITT| 69.5| 75.34 ± 0.18 | 76.18 ± 0.16|
> | | replace TOT | 60.5 | 60.10 ± 0.11| 57.82 ± 0.17 |
> | | swap ITT | 45.7 | 61.55 ± 0.38 | 61.60 ± 0.07|
> | | swap TOT | 25.9 | 27.86 ± 0.08 | 23.34 ± 0.19|
> | **SugarCrepe** | add ITT | 72.9| 84.56 ± 0.09| 86.27 ± 0.15 |
> | | replace ITT | 80.0| 83.84 ± 0.14| 84.80 ± 0.04|
> | | swap ITT| 62.7| 73.79 ± 0.12| 72.54 ± 0.04|
> | **Wino** |txt score| 31.2| 31.75 ± 0.20| 32.00 ± 0.61 |
> || img score| 11.0| 11.83 ± 0.31| 11.67 ± 0.12 |
> || group score| 8.7| 9.33 ± 0.31 | 10.08 ± 0.24 |
> | **COCO** | img ret.| 54.6 | 60.32 ± 0.64 | 59.42 ± 0.04 |
> || txt ret.| 74.1 | 75.92 ± 0.17| 76.00 ± 0.09 |
> | **Flickr30K** | img ret. | 83.5| 86.70 ± 0.03 | 86.27 ± 0.02 |
> || txt ret. | 95.1| 95.10 ± 0.08 | 95.67 ± 0.05 |
> | **ImageNet** | acc | 63.3 | 61.72 ± 0.02 | 62.41 ± 0.09 |
> | **ZS10** | acc| 61.4| 60.89 ± 0.06| 60.22 ± 0.03 |
>
> [1] Zhang, Beichen, et al. "Long-clip: Unlocking the long-text capability of clip." ECCV, 2024.

---

> > ### Author Response · Authors · 2025-08-08
> >
> > Thanks a lot for your time and the positive review. We would be happy to answer any further questions in the remaining time.

---

### Note · Authors · 2025-08-15

We thank all reviewers for the constructive discussion and will integrate the additional results of the rebuttal into the paper. All of the reviewers’ queries have been addressed, and we summarize the main points.

We introduce CLIC, a computationally efficient fine-tuning method to enhance compositionality in CLIP models. CLIC concatenates two images and two sentences of their captions to create:

   **Rich positives** by combining sentences in different order and noting that each sentence in a caption describes different aspects of the images, we get grammatically correct positive captions which have semantic and lexical variability

   **Hard negatives** by swapping words of the same category between captions of two images, we create hard negatives, since the entities/actions/attributes are present in the two images but in a wrong relationship (2CLt)

CLIC trains the CLIP model to (1) distinguish captions that are lexically similar but semantically different, and (2) align captions that are lexically different but semantically similar.

**Effectiveness**: Achieves SOTA compositionality (SugarCrepe++), **improves** retrieval, and only slightly degrades zero-shot classification - unlike most prior methods, which heavily degrade retrieval and classification. CLIC based learning is very fast (only 40 min for ViT-B-32 for fine-tuning on 1M Laion samples with 4 GPUs).

**Generality and reproducibility**: Works across multiple CLIP architectures (ViT-B-32, ViT-B-16, ViT-L-14), pretraining datasets and methods (OpenAI, CLIPS, CLIPA, etc.), finetuning datasets (Laion, RedCAPS, CC12M, MS-COCO), and different tokenizers (OpenAI, CLIPS). Multiple runs yield low variance in results (m5ip).

**Adaptability and robustness to grammatical cues (m5ip, 2CLt)**: We clarified
that potential ungrammatical captions are only used as negatives, and we see no bias due to abrupt topic changes, as evidenced by improving results on long-caption retrieval. Moreover, non-grammatical negatives work as well as grammatically
correct ones generated via LLMs. CLIC can be adapted for structured, hierarchical compositionality by pairing images containing the same objects but with differing attributes.

**VLMs**: While in the paper we fine-tune only the text-encoder, CLIC fine-tuning of text and vision encoder and replacing the vision encoder in Llava 1.5 improves compositionality while maintaining utility (PGUp). In the final paper, we show results for a fully-trained Llava model.

---

### Decision · Program_Chairs · 2025-09-17

**Decision:**

Accept (poster)

**Comment:**

This paper introduces CLIC, a simple and computationally efficient fine-tuning method to improve the compositional reasoning of vision-language models like CLIP. The core idea is to create challenging training examples by concatenating two image-caption pairs. This process is used to generate a diverse set of positive examples and hard-negative examples. The authors demonstrate that this method significantly improves performance on compositional benchmarks, particularly on the challenging SugarCrepe++ dataset, while maintaining or even enhancing performance on downstream retrieval and classification tasks.

The reviewers unanimously agree that the paper addresses a significant and well-known limitation in vision-language models. There were some concerns in the initial reviews, but authors addressed these concerns. The final review scores are three borderline accepts and an accept. Therefore, we have decided to accept this paper.